# Chemical genetic identification of GAK substrates reveals its role in regulating Na⁺/K⁺-ATPase

Amy W Lin[1],*, Kalbinder K Gill[1],*, Marisol Sampedro Castañeda[1], Irene Matucci[1], Noreen Eder[1,2], Suzanne Claxton[1], Helen Flynn[2], Ambrosius P Snijders[2], Roger George[3], Sila K Ultanir[1]

Cyclin G–associated kinase (GAK) is a ubiquitous serine/threonine kinase that facilitates clathrin uncoating during vesicle trafficking. GAK phosphorylates a coat adaptor component, AP2M1, to help achieve this function. GAK is also implicated in Parkinson's disease through genome-wide association studies. However, GAK's role in mammalian neurons remains unclear, and insight may come from identification of further substrates. Employing a chemical genetics method, we show here that the sodium potassium pump (Na⁺/K⁺-ATPase) α-subunit Atp1a3 is a GAK target and that GAK regulates Na⁺/K⁺-ATPase trafficking to the plasma membrane. Whole-cell patch clamp recordings from CA1 pyramidal neurons in GAK conditional knockout mice show a larger change in resting membrane potential when exposed to the Na⁺/K⁺-ATPase blocker ouabain, indicating compromised Na⁺/K⁺-ATPase function in GAK knockouts. Our results suggest a modulatory role for GAK via phosphoregulation of substrates such as Atp1a3 during cargo trafficking.

## Introduction

Cyclin G–associated kinase (GAK), also known as auxilin 2, is a member of the Numb-associated kinase (NAK) family. Other members include adaptor-associated kinase 1 (AAK1), BMP2-inducible kinase (BIKE/BMP2), and myristoylated and palmitoylated serine/threonine kinase 1 (MPSK1) (Smythe & Ayscough, 2003; Sorrell et al, 2016). GAK is ubiquitously expressed in various tissues (Kanaoka et al, 1997; Kimura et al, 1997) and is primarily known for its role in clathrin uncoating (Zhang, Engqvist-Goldstein et al, 2005b; Lee et al, 2005; Lee et al, 2006). At its N-terminus, GAK has a highly conserved serine/threonine kinase domain followed by a lipid-binding PTEN-like domain (Lee et al, 2006), a clathrin-binding domain (Kametaka et al, 2007), and a J-domain, the latter two domains being sufficient for Hsc70 and GAK-mediated uncoating of clathrin-coated

vesicles (Greener et al, 2000; Umeda et al, 2000). Conventional GAK knockout in mice is embryonic lethal, while conditional knockout of GAK in the developing brain, liver, and skin all result in death shortly after birth (Lee et al, 2008). Knockout of GAK in adult mice using a tamoxifen-inducible Cre recombinase also resulted in death (Lee et al, 2008). A transgenic mouse expressing only the clathrin-binding and J-domains was able to rescue the lethality of GAK knockout in the liver and brain, showing that neither the PTEN-like domains nor the kinase domain was necessary for GAK function in these tissues (Park et al, 2015). However, kinase-dead GAK knock-in mice still led to neonatal lethality caused by defective lung development (Tabara et al, 2011), indicating importance for GAK activity. As a serine/threonine kinase, GAK exerts part of its functional effects by targeting downstream substrates. The only currently known substrates of GAK are the μ2- and μ1-subunits of the adaptor proteins AP-2 and AP-1, respectively (Umeda et al, 2000; Korolchuk & Banting, 2002; Kametaka et al, 2007), and protein phosphatase 2A (Naito et al, 2012). However, the functional role of GAK is still not fully understood in neurons, and a number of unidentified GAK substrates likely remain.

The sodium potassium pump (Na⁺/K⁺-ATPase, NKA) is a transmembrane ion pump critical in the maintenance of resting membrane potential (RMP). It utilizes the hydrolysis of an ATP molecule to drive the extrusion of three Na⁺ ions from the cell and uptake of two K⁺ ions into the cell against their electrochemical gradients. Structurally, it is a heteromer consisting of three subunits: α, β, and γ. In mammals, there are four catalytic α-subunits, α1–4. The α1–3-subunits are expressed in brain tissue at varying levels, with α1 being ubiquitous, α2 mainly glial, and α3 neuronal (Bottger et al, 2011). There are three β-subunits: the β1-subunit is expressed ubiquitously, the β2 isoform is expressed in brain and muscle (Blanco, 2005), and the β3 isoform is mainly expressed in lung, testis, skeletal muscle, and liver (Malik et al, 1996; Arystarkhova & Sweadner, 1997). In some tissues, there is also an additional modulatory γ subunit, the FXYD protein, of which there are seven (Sweadner & Rael, 2000; Crambert & Geering, 2003). The NKA undergoes a number of posttranslational modifications that

[1]Kinase and Brain Development Lab, The Francis Crick Institute, London, United Kingdom    [2]Mass Spectrometry Platform, The Francis Crick Institute, London, United Kingdom    [3]Protein Purification Facility, The Francis Crick Institute, London, United Kingdom

Correspondence: Sila.Ultanir@crick.ac.uk
*Amy W Lin and Kalbinder K Gill contributed equally to this work.

affect its trafficking and function, including ubiquitination (Coppi & Guidotti, 1997; Thevenod & Friedmann, 1999) and phosphorylation (Poulsen et al, 2010). Mutations in *ATP1A3* are known to cause rapid-onset dystonia parkinsonism (de Carvalho Aguiar, Sweadner et al, 2004; Bottger et al, 2011), as well as alternating hemiplegia of childhood and CAPOS (cerebellar ataxia, areflexia, pes cavus, optic atrophy, and sensorineural hearing loss) syndrome (Clausen et al, 2017).

Identification of direct kinase substrates can reveal previously unknown molecular and cellular kinase functions, which is critical for disease-relevant pathway expansion where kinase–disease genetic links are well established. Direct kinase substrate identification is a challenging problem in cell signalling. In this study, we engineered the kinase domain of GAK to enable its use of bulky ATP analogs with γ thiophosphates. Analog-specific GAK was then used to label its substrates specifically. These substrates and their phosphorylation sites were identified with mass spectrometry, revealing putative novel GAK substrates. Using phosphospecific antibodies, we validated Atp1a3, the catalytic α3-subunit of NKA, and Sipa1l1 (signal-induced proliferation–associated 1-like protein 1) as novel GAK substrates. Using overexpression studies in neurons, COS7, and human embryonic kidney (HEK) 293T cells, we show that Atp1a3 T705 is phosphorylated by GAK and another NAK family kinase AAK1, and it is important for NKA localization and function. In order to further study GAK function in neurons, we crossed GAK-floxed (F) mice with neuronal Cre lines to obtain conditional GAK knockout mouse models. We show that GAK knockout neurons were more sensitive to NKA inhibition by ouabain, indicating alterations in NKA function. Our results describe a novel role for GAK in regulating Atp1a3 function.

## Results

### GAK protein engineering and site suppressor rescue mutations

To identify downstream targets of GAK in brain tissue, we used a chemical genetic method of labelling kinase substrates (Blethrow et al, 2004; Hertz et al, 2010). To do so, we first had to engineer a pocket in the GAK protein. Using a truncated form of GAK from aa 1–400 (GAK$^{1–400}$) containing the GAK domain (aa 54–311), we generated two ATP analog–specific (AS) mutants, T123G and T123A, by mutating the hydrophobic "gatekeeper" residue of the GAK ATP-binding pocket to a smaller aa (Fig 1A). This enlarges the ATP-binding pocket and allows ATP analog specific mutants to utilize bulky ATP-γ-S analogs containing a thiol group to selectively thiophosphorylate their substrates (Fig 1B). Thiophosphorylated substrates can be detected via Western blot after alkylation by p-nitrobenzylmesylate (PNBM) using an anti-thiophosphate ester antibody (Allen et al, 2007). To determine which bulky ATP-γ-S analog was most compatible with GAK mutant kinases, we performed an in vitro kinase reaction using GAK autophosphorylation as a readout of GAK activity. We found that GAK T123A could use furfuryl-ATP-γ-S, but at reduced levels (Fig 1C), showing GAK to be an intolerant kinase. To rescue this loss, we needed to identify potential secondary-site suppressor mutations. We mutated the aa

residue preceding the conserved "DFG" motif, C190, to alanine or threonine, as the site was previously reported to rescue the activity of intolerant kinase NDR1 (Ultanir et al, 2012). We also introduced mutations at a second aa site that comes into close contact with the N-6 position of ATP adenine (Eblen et al, 2003), V99 in GAK. However, we observed no changes in GAK activity with any of these strategies (Fig S1). We next took a closer look at the structure of GAK in comparison with other kinases. A key component known to affect kinase activity is the regulatory spine, which consists of four hydrophobic residues spanning the tertiary structure of the kinase domain (Kornev et al, 2006). The introduction of isoleucine, a hydrophobic β-branched aa, at a −2 position with respect to the gatekeeper residue located on the β5 strand (Fig 1D), is able to compensate for the loss of hydrophobicity caused by gatekeeper mutation (Joseph & Andreotti, 2011). In GAK, the hydrophobic spine residues are L89, F101, H171, and F192 (Fig 1D). We therefore introduced a L121I mutation to the GAK T123A mutant and observed that L121I moderately rescued kinase activity (Fig 1E). Finally, we made use of GAK structure–based sequence alignment with the other NAK family members (Zhang, Kenski et al, 2005a) using mouse protein sequences and noticed that the first hydrophobic spine residue L89 was a methionine in all other members, as well as in the human GAK sequence. To mimic AAK1's sequence in that region, we introduced a C87Q/F88I/L89M mutation to the L121I/T123A GAK mutant (Fig 1A and D). We found that these rescue mutations together (AS) restored GAK activity towards furfuryl-ATP-γ-S to levels comparable with GAK WT using ATP-γ-S (Fig 1E). We used this final combination of mutations T123A/L121I/C87Q/F88I/L89M in GAK$^{1–400}$ (referred to as GAK AS) during subsequent experiments to thiophosphorylate putative substrates in mouse brain tissue.

### Chemical genetic identification of GAK substrates

Before performing covalent capture, we first wanted to confirm and characterize the presence of GAK in the wildtype mouse brain. GAK is expressed from postnatal day (P)4–P50 (Fig S2A) throughout all brain regions (Fig S2B).

In order to perform covalent capture for kinase substrate identification, GAK$^{1–400}$ was first purified from insect cells, as full length (FL) GAK was not stable enough to purify. GAK AS was purified along with a K69R kinase dead (KD) mutant of GAK$^{1–400}$ (GAK KD) as a negative control. P11 mouse brain lysates were then incubated with furfuryl-ATP-γ-S and either purified GAK KD or GAK AS to label potential substrates. Afterwards, brain lysate proteins were digested with trypsin. Any thiol-containing peptides (Fig 2A, red), including those with cysteine aas (Fig 2A, blue), were captured with iodoacetyl agarose beads, while peptides phosphorylated by endogenous kinases (Fig 2A, grey) were washed out. Using oxone-induced hydrolysis, cysteine-linked peptides were left on the beads while thiophosphate ester–linked peptides were released as phosphopeptides (Fig 2A, red). These eluted peptides were then subjected to liquid chromatography/tandem mass spectrometry analysis to identify substrates and their phosphorylation site (Fig 2A).

From the data generated, only phosphopeptides detected at least three times in GAK AS samples and not in any GAK KD negative controls are included in a list of substrate candidates. Of the

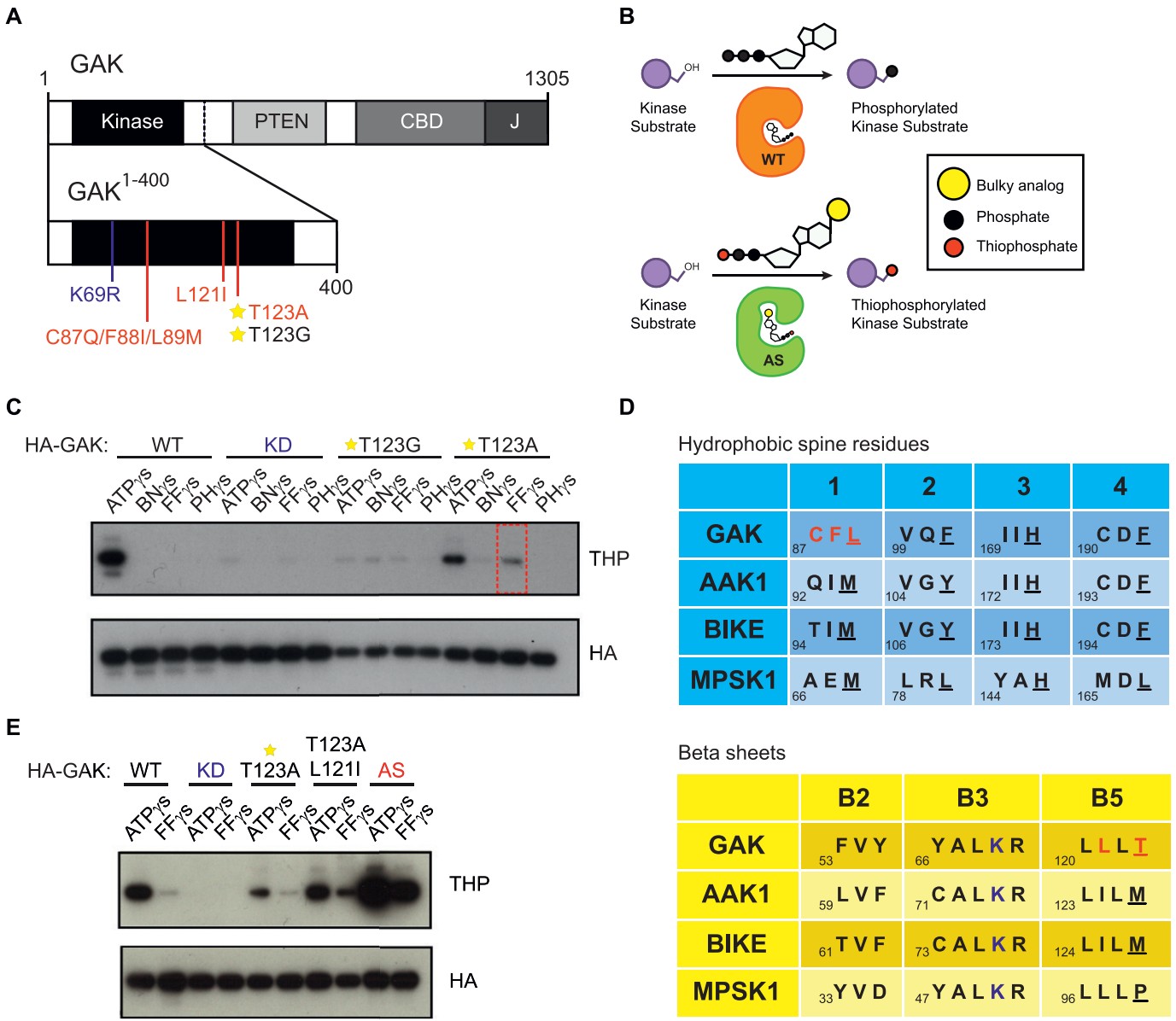

**Figure 1. GAK protein engineering and site suppressor rescue mutations.**
**(A)** A diagram showing both FL GAK with its major domains and GAK[1–400] with various mutation sites. The KD mutation is shown in blue, the gatekeeper Thr site with a star, and the AS mutations in red. **(B)** A cartoon illustrating the chemical genetics strategy of using a gatekeeper mutation to enlarge the ATP-binding pocket of a kinase to allow detection via a non-radioactive method. **(C)** An initial kinase screen of HA-tagged GAK gatekeeper mutants T123A and T123G with bulky ATP analogs (BN, Benzyl; FF, Furfuryl; PH, Phenyl) shows that although T123A can use FF (red box), kinase activity is greatly reduced. Thiophosphate ester antibody (THP) is used as a readout of kinase activity. **(D)** Kinase alignment of NAK family hydrophobic spine residues and β sheets as a strategy to identify potential secondary site suppressor mutations. Red indicates the aas mutated in the final AS kinase. Blue indicates the active lysine site. In the hydrophobic spine table, hydrophobic spine residues are underlined. In the β-sheet alignment table, the gatekeeper residue is underlined. **(E)** Site suppressor mutations can rescue the loss of GAK activity caused by gatekeeper mutation. The final AS mutation containing T123A/L121I/C87Q/F88I/L89M successfully restores GAK activity.

potential GAK substrates identified, there appears to be a predominant preference for a small TG motif, where T is phosphorylated threonine and G is glycine (Fig 2A). This motif matches with that of a previously identified GAK and AAK1 substrate, the T156 residue of the AP-2 adaptor complex μ2-subunit (SQITSQV**TG**-QIGWRR) (Korolchuk & Banting, 2002; Ricotta et al, 2002; Zhang et al, 2005b), as well as the GAK substrate protein phosphatase 2A (PP2A) B'γ T104 residue (SNP**TG**AEFDP) (Naito et al, 2012). Therefore, in the

list of putative GAK substrates, we removed the three candidates that do not contain the "TG" motif (Table 1). The dataset was uploaded in PRIDE # PXD011319. Neither AP2 μ2 nor PP2A B'γ appeared in our list of candidate substrates, which could be due to either low abundance or that using GAK[1–400] instead of FL GAK resulted in less effective interaction with binding partners. In addition, due to their biophysical properties, some peptides may not be detected by mass spectrometry. Therefore, not detecting a

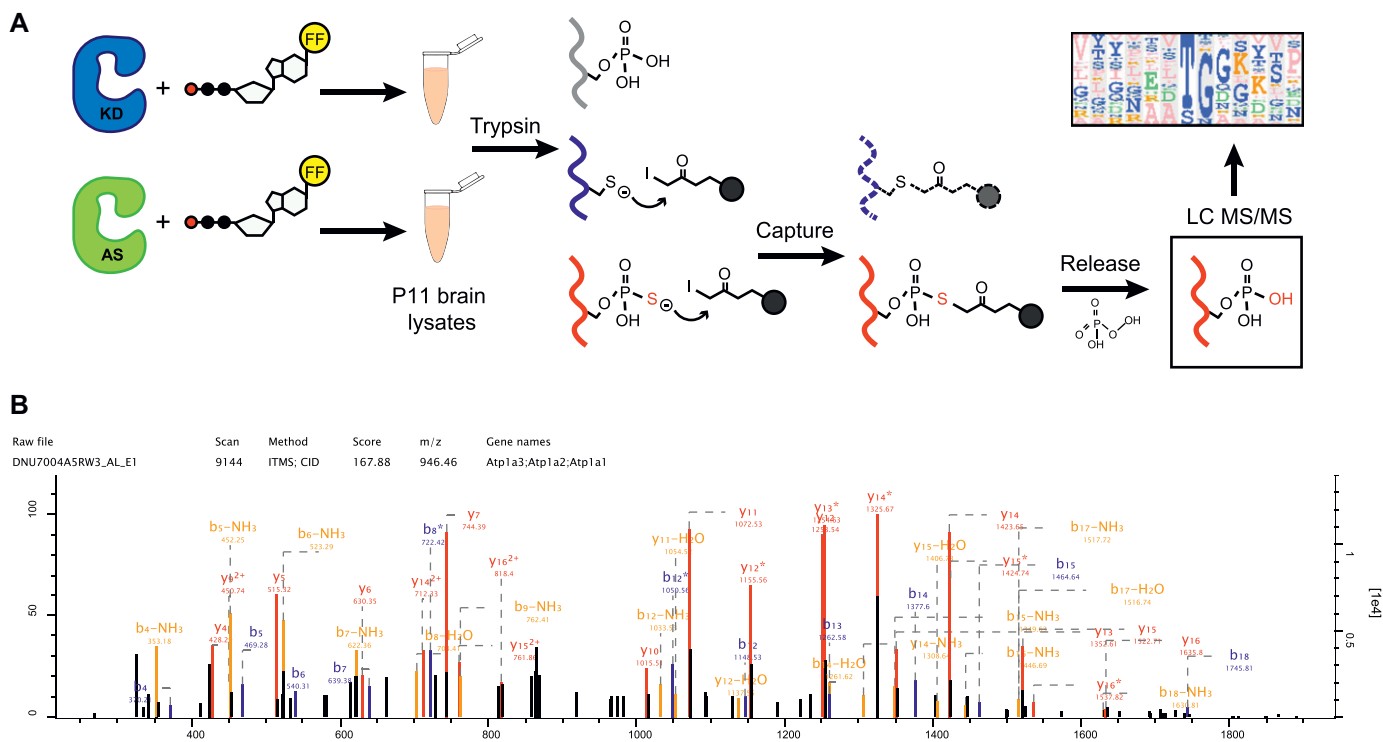

**Figure 2. Chemical genetic identification of GAK substrates.**
**(A)** A cartoon showing the covalent capture method for identification of kinase substrates from P11 mouse brain lysates. Grey represents proteins phosphorylated by endogenous kinases. Red shows thiophosphorylated proteins and blue indicates cysteine-containing proteins. **(B)** GAK substrates Atp1a3 and Sipa1L1 are identified during a chemical genetics screen. Several other potential candidates are shown to illustrate the presence of a small TG motif. * Indicates the phosphorylated threonine residue identified. Bold indicates the small TG motif consistent with identified substrates. **(C)** Mass spectroscopy identification of Atp1a3 phosphorylation by spectra analysis of Atp1a3-derived peptides containing phosphorylated T705.

phosphorylation site would not necessarily mean that it is not phosphorylated. The putative substrates themselves appear to reflect the ubiquitous nature of GAK (Table 1). We chose to further study two of these substrates due to their important functions in neurons: Sipa1L1, which regulates dendritic spine morphology, and the catalytic α-subunits of the sodium potassium pump

**Table 1. Table of potential GAK substrates.**

| Position | Leading proteins | Protein names | Gene names | Sequence window |
|---|---|---|---|---|
| T425 | Q9QWI6 | SRC kinase signalling inhibitor 1 | Srcin1 | GSPVHHAAERL**GGAPTGQ**GVSPSPSAILERR |
| T55 | Q9D6F9 | Tubulin β-4A chain | Tubb4a | SDLQLERINVYYNEA**TG**GNYVPRAVLVDLEP |
| T472 | O88532 | Zinc finger RNA-binding protein | Zfr | NTSSIATSSVKGLST**TG**NSSLNSTSNTKVSA |
| T249 | Q8C0T5 | Signal-induced proliferation–associated 1-like protein 1 | Sipa1l1 | SDRGPTPTKLSDFLI**TG**GGKGSGFSLDVIDG |
| T385 | Q3UIZ0 | Cyclin-GAK | Gak | GGFLDILRGGTERLF**T**NLKDTSSKVIQSVAN |
| T705 | Q8VCE0 | Sodium/potassium-transporting ATPase subunit α-3; α-1; α-2 | Atp1a3;Atp1a1;Atp1a2 | IIVEGCQRQGAIVAV**TG**DGVNDSPALKKADI |
| T3 | Q64213 | Splicing factor 1 | Sf1 | ___________MA**TG**ANATPLDFPSKKRK |
| T55 | P68372 | Tubulin β-4B chain | Tubb4b;Tubb2c | SDLQLERINVYYNEA**TG**GKYVPRAVLVDLEP |
| T118 | P60202 | Myelin proteolipid protein | Plp1 | DYKTTICGKGLSATV**TG**GQKGRGSRGQHQAH |
| T465 | A2ARP8 | Microtubule-associated protein 1A;MAP1 light chain LC2 | Mtap1a;Map1a | MQFLMQKWAGNSKAK**TG**IVLANGKEAEISVP |
| T627 | Q3UR70 | Transforming growth factor-β receptor–associated protein 1 | Tgfbrap1 | LAILYLEEVLRQRVS**TG**GKDVEATETQAKLR |

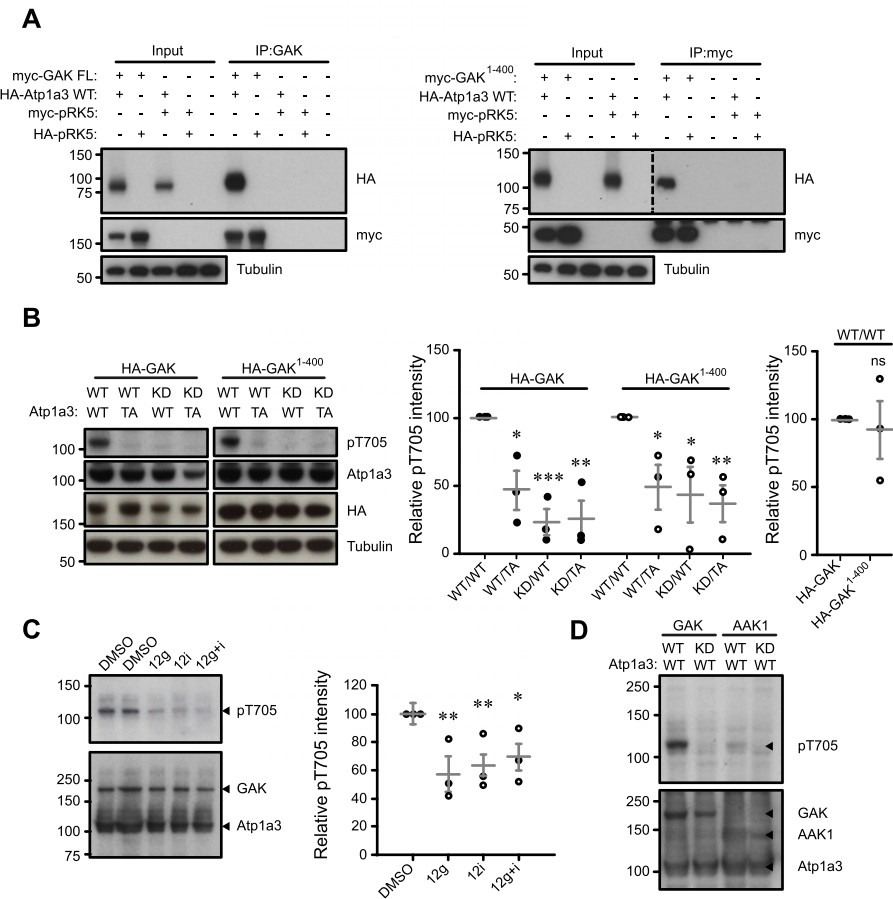

**Figure 3.  GAK phosphorylates Atp1a3 at novel site T705 in HEK 293Ts.**
**(A)** Atp1a3 is immunoprecipitated by GAK from HEK 293T cells co-transfected with myc-tagged FL GAK WT and HA-tagged Atp1a3 WT (left panel). Atp1a3 is also able to be immunoprecipitated by myc when cells are co-transfected with myc-tagged GAK[1–400] and HA-tagged Atp1a3 WT (right panel). HA-tagged and myc-tagged pRK5 vectors have been used as IP controls. **(B)** Atp1a3 pT705 is present in HEK 293T samples co-transfected with Atp1a3 WT and either FL GAK WT or GAK[1–400] (left). Atp1a3 T705A mutants show reduced pT705, as do GAK KD mutants (centre). GAK[1–400] pT705 levels are similar compared to FL GAK (right). Graph data represent means ± SEM, $n = 3$ independent experiments, GAK FL (WT/TA $P = 0.02$, KD/WT $P = 0.001$, KD/TA $P = 0.005$), GAK[1–400] (WT/TA $P = 0.04$, KD/WT $P = 0.05$, KD/TA $P = 0.009$), *$P < 0.05$, **$P < 0.01$, ***$P < 0.001$, $t$ test. **(C)** HEK 293T cells co-transfected with HA-tagged FL GAK WT and HA-Atp1a3 WT and pretreated in serum-free media were incubated with either DMSO or 5 $\mu$M of GAK inhibitor 12g, 12i, or 12g+i for 1 h $n = 3$ independent in duplo experiments. Graph data represent means ± SEM, 12g $P = 0.009$, 12i $P = 0.009$, 12g+i $P = 0.03$; *$P < 0.05$, **$P < 0.01$, $t$ test. **(D)** Co-transfections of HEK 293T cells with HA-tagged Atp1a3 WT and either HA-tagged GAK or AAK1 demonstrate that GAK is the primary kinase phosphorylating Atp1a3 at T705. $n = 5$ independent experiments.

(Atp1a1/Atp1a2/Atp1a3), which maintain RMP. The phosphorylation site we identified was conserved among α-subunit subtypes; as Atp1a3 is the most abundant α subunit in neurons, we focused our attention to Atp1a3.

## GAK phosphorylates Sipa1L1 at T249 in HEK 293T cells

Sipa1L1, also known as SPAR, was first identified as a GTPase-activating protein for Rap that was targeted to dendritic spines, where it induced reorganization of F-actin filaments to promote dendritic spine head growth (Pak et al, 2001). It is a large, multi-domain protein containing two separate regions (Act1 and Act2) capable of independently reorganizing F-actin (Fig S3A) (Pak et al, 2001). From our GAK substrate screen, we identified the T249 site, located in the Sipa1L1 Act1 domain (aa 1–627) as the site of GAK phosphorylation (Fig S3A). The T249 site is conserved between mouse, rat, and human Sipa1L1, but is not conserved between Sipa1L1 and its isoforms (Sipa1L2 and Sipa1L3). In order to study Sipa1L1 phosphorylation, we generated an Act1 domain–only form of Sipa1L1 (Act1) (Fig S3A). Firstly, we tested whether GAK and Sipa1L1 are able to interact by co-expressing myc-tagged GAK FL with either HA-tagged Sipa1L1 or HA-tagged Act1 in HEK 293T cells and immunoprecipitating GAK. We find that GAK and Sipa1L1 do interact and that Act1 can coimmunoprecipitate with GAK (Fig S3B). To test whether or not GAK phosphorylates Sipa1L1 at T249, we generated a

rabbit polyclonal phosphospecific antibody against Sipa1L1 T249 (pT249) and made a non-phosphorylatable Sipa1L1 Act1 domain T249A mutant (TA). We then overexpressed GAK FL WT or GAK FL KD with either Act1 WT or TA in HEK 293T cells. Our results show that GAK FL WT can phosphorylate Sipa1L1 at T249 while GAK FL KD cannot, confirming it as a substrate in cells (Fig S3C). We next examined endogenous pT249 expression in neural tissue and found that Sipa1L1 pT249 was present in both embryonic and adult rat brain tissue but was higher in embryonic brains, indicating possible developmental regulation (E18 pT249 53.9 ± 4.8; adult pT249 11.9 ± 3.4; Fig S3D). The amount of total Sipa1L1 was lower in E18 than in adult brains (E18 Sipa1L1 16.1 ± 1.6; adult Sipa1L1 92.6 ± 8.4; Fig S3D), and GAK levels were fairly similar though slightly elevated in adult brains (E18 GAK 106 ± 7.9; adult GAK 123 ± 7.5; Fig S3D). These results confirm that Sipa1L1 is a substrate of GAK in cells.

## GAK phosphorylates Atp1a3 at novel site T705 in HEK 293T cells

We identify a novel phosphorylation site at T705 of the NKA α-subunit (Fig 2B). The region encompassing T705 is conserved between all four α-subunit isoforms as well as between mice, rats, and humans. As α3 is a neuronal subunit (Bottger et al, 2011), we focused our work on the α3 isoform. We reasoned that if Atp1a3 is phosphorylated by GAK, the proteins may demonstrate some form of interaction. To test whether these two proteins are able to interact,

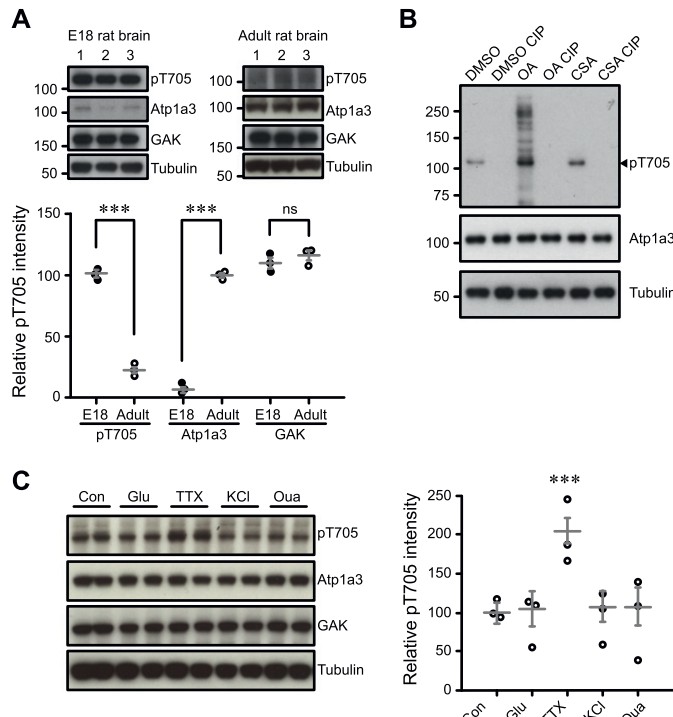

**Figure 4. Endogenous phosphorylation of Atp1a3 is regulated by phosphatases.**
**(A)** pT705 is present in E18 rat embryonic brain lysates but decreased in adult rat brain lysate ($P = 3.4 \times 10^{-5}$), while Atp1a3 is low in E18 and elevated in adult brain ($P = 9.1 \times 10^{-6}$). GAK levels are relatively similar ($P = 0.35$). Graph data represent means ± SEM, $*P < 0.05$, $**P < 0.01$, $***P < 0.001$, $t$ test. Numbers 1–3 represent independent samples from different brains. **(B)** Cortical rat cultures treated with phosphatase inhibitors OA (0.5 $\mu$M) and CSA (10 $\mu$M) show an increase in pT705 intensity that is absent after CIP treatment. **(C)** Rat cultured hippocampal neurons were treated with the following for 30 min: no treatment control (Con), 50 $\mu$M glutamate (Glu) $P = 0.86$, 1 $\mu$M tetrododoxin (TTX) $P = 0.0002$, 50 mM KCl $P = 0.74$ or 30 $\mu$M ouabain (Oua) $P = 0.77$. $n = 3$ independent in duplo experiments. $*P < 0.05$, $**P < 0.01$, $***P < 0.001$, $t$ test.

we co-expressed myc-tagged FL GAK with HA-tagged Atp1a3 in HEK 293T cells and immunoprecipitated GAK to confirm pull down of Atp1a3 (Fig 3A, left panel). We also tested to see whether myc-tagged GAK[1–400] was able to co-immunoprecipitate HA-tagged Atp1a3. Here, we immunoprecipitated with myc instead of GAK due to the truncation of the C-terminus and found that the GAK domain is able to pull down Atp1a3 (Fig 3A, right panel).

To validate Atp1a3 T705 as a GAK phosphorylation site, we generated a rabbit polyclonal phosphospecific antibody against Atp1a3 T705 (pT705). We made an Atp1a3 phosphomutant, Atp1a3 T705A (TA), which cannot be phosphorylated, and then overexpressed FL GAK wildtype (GAK FL WT) or KD (GAK FL KD) with either Atp1a3 wildtype (WT) or Atp1a3 TA in HEK 293T cells. To determine whether the kinase domain alone is sufficient to phosphorylate Atp1a3 T705, we also overexpressed GAK[1–400] WT or KD (GAK[1–400] KD) with either Atp1a3 WT or Atp1a3 TA. Our results suggest that Atp1a3 T705 is phosphorylated by GAK and that the GAK domain is as sufficient as FL GAK (WT/WT HA-GAK[1–400] 92.8 ± 21.3; Fig 3B, right). We observed a large reduction in pT705 levels in cells co-transfected with the KD form of GAK (KD/WT 22.7 ± 9.6; Fig 3B, centre) and GAK[1–400] (KD/WT 42.9 ± 20.4; Fig 3B, centre) when normalized to their

respective WT/WT controls. This reduction was also seen with the Atp1a3 TA mutant for both GAK FL (WT/TA 46.2 ± 14.4; KD/TA 25 ± 13.5; Fig 3B, centre) and GAK[1–400] (WT/TA 48.3 ± 16.5; KD/TA 36.3 ± 13.5; Fig 3B, centre). To further confirm that GAK activity causes Atp1a3 T705 phosphorylation, we treated HEK 293T cells co-transfected with GAK FL WT and Atp1a3 WT with GAK inhibitors 12g IVAP-1966 and/or GAK inhibitor 12i IVAP-1967 (5 $\mu$M each) for 1 h under serum-free conditions (Kovackova, Chang et al, 2015a). Both GAK inhibitors reduced Atp1a3 T705 phosphorylation (12g 57.5 ± 12.9, 12i 63.3 ± 7.47, 12g+i 69.1 ± 9.26) compared to DMSO (100 ± 7.73) (Fig 3C). However, these GAK inhibitors do not fully eliminate phosphorylation of T705. It is a possibility that other members of the NAK family might also target Atp1a3 T705. For example, both AAK1 and GAK phosphorylate AP-2 $\mu$2 (Korolchuk & Banting, 2002; Lee et al, 2005; Ricotta et al, 2002). To examine whether AAK1 is able to phosphorylate Atp1a3 at T705, we co-transfected Atp1a3 WT or Atp1a3 TA with either AAK1 WT or AAK1 KD and then ran an immunoblot to examine Atp1a3 pT705 levels. We find that while AAK1 WT can phosphorylate Atp1a3 WT, the level of phosphorylation by AAK1 WT is much weaker compared to GAK WT (Fig 3D). These experiments show that GAK, and possibly other NAK family members, can phosphorylate Atp1a3 at T705 in cells. Interestingly, mutations in the *ATP1A3* gene are known to cause rapid-onset dystonia parkinsonism (de Carvalho Aguiar et al, 2004). Considering the critical role of Atp1a3 in neuronal function, we decided to investigate Atp1a3 further.

## Endogenous phosphorylation of Atp1a3 is regulated by phosphatases

We next examined endogenous pT705 levels in neural tissue. Endogenous pT705 is readily detectable in embryonic day (E)18 rat brains (E18 pT705 101 ± 2.3; Fig 4A) despite lower expression of total Atp1a3 in rat embryos compared to adult rat (Orlowski & Lingrel, 1988) (E18 Atp1a3 6.7 ± 2.7; adult Atp1a3 99.8 ± 1.9; Fig 4A). However, the amount of pT705 in adult rats (adult pT705 22.6 ± 3.1; Fig 4A) appears lower despite increased abundance of total Atp1a3 (Fig 4A), which might suggest a potential developmental role for Atp1a3 T705 phosphorylation. GAK protein levels are detectable in fairly similar amounts in adult rat brain compared to embryonic brain (E18 GAK 109.7 ± 4.4; adult GAK 116 ± 4.0; Fig 4A).

To test the phosphatase-dependent regulation of pT705, we treated rat primary cortical cultures with phosphatase inhibitors okadaic acid (OA), cyclosporin A (CSA) and DMSO vehicle. Phosphorylation levels were increased upon phosphatase inhibitor treatment, most notably OA. To ensure that our antibody is phosphospecific, we dephosphorylated either DMSO-, OA- or CSA-treated samples by calf alkaline phosphatase (CIP) treatment of protein lysates. CIP-treated samples clearly demonstrated the loss of pT705 (Fig 4B, lanes 2, 4, 6), while total Atp1a3 levels remained constant. These experiments show that endogenous pT705 phosphorylation is present in neurons and is negatively regulated by phosphatases PP1, PP2A, and PP2B (calcineurin). Our results indicate that Atp1a3 T705 phosphorylation occurs in cultured neurons and in vivo.

Atp1a3 function is important for maintaining RMP after large rapid sodium influxes such as those observed during action

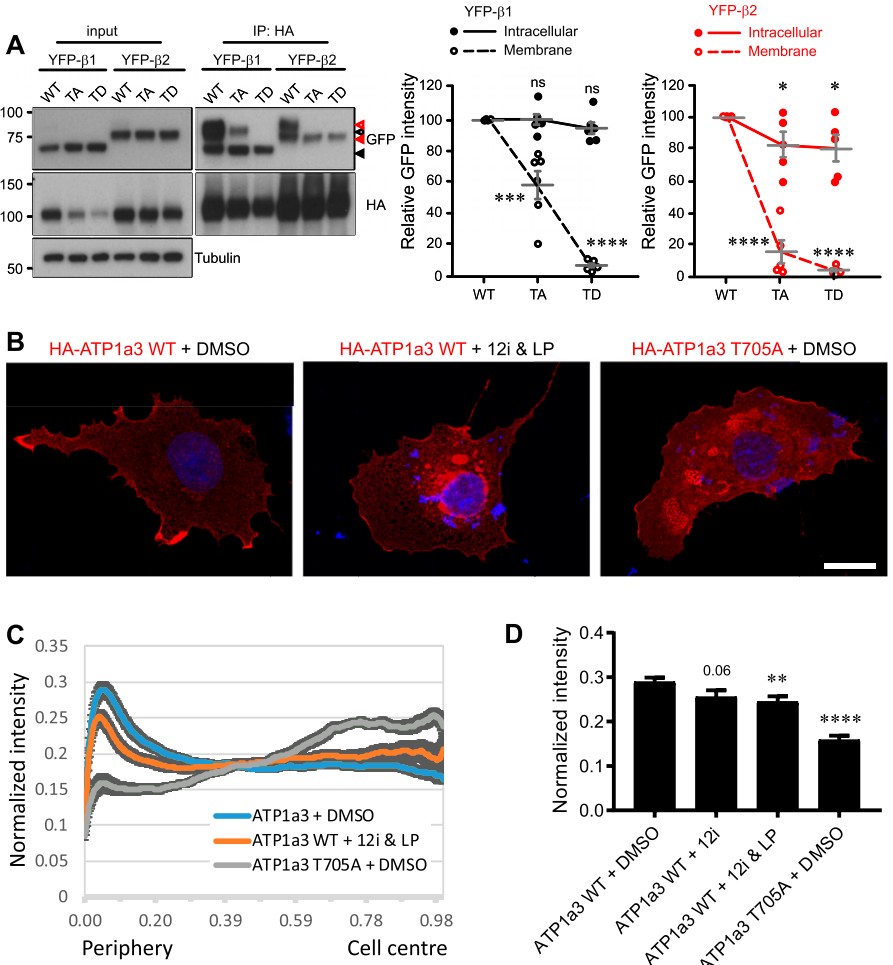

**Figure 5. Atp1a3 phosphomutants show alterations in subcellular localization.**
**(A)** Co-transfections of either YFP-β1 (black) or YFP-β2 (red) with HA-tagged Atp1a3 WT, TA, and TD in HEK 293T cells show different immunoprecipitated glycosylated forms of the ATPase β-subunit. Solid arrows indicate lower MW intracellular forms, and empty arrows indicate the higher MW plasma membrane forms of the β-subunit. Graph data represent means ± SEM, $n$ = 5 independent experiments, TA/β1 intra $P$ = 0.0009, TD/β1 intra $P$ = 6.2 × $10^{-9}$, TA/β2 intra $P$ = 2.5 × $10^{-6}$, TA/β2 membrane $P$ = 0.05, TD/β2 intra $P$ = 4.4 × $10^{-13}$, TD/β2 membrane $P$ = 0.04; *$P$ < 0.05, **$P$ < 0.01, ***$P$ < 0.001, ****$P$ < 0.0001, $t$ test. **(B)** COS7 cells transfected with HA-tagged Atp1a3 WT with DMSO treatment as negative control, dual NAK family inhibitor treatment 12i IVAP-1967 (12i) + LP-935509 (LP) and Atp1a3 T705A mutant cellular distribution are shown (red is HA immunostaining showing Atp1a3, blue is DAPI). Scale bar = 25 μm. **(C)** Atp1a3 staining intensity was normalized to total intensity value for each cell. Y axis shows normalized distance from cell's centre (1) to its periphery (0). The plots show average and standard error of mean ($n$ = 23, 34, 31 cells for Atp1a3 + DMSO, Atp1a3 + 12i + LP, Atp1a3 T705A, respectively). **(D)** Data points at the peak for the control group at periphery are compared using ANOVA and paired $t$ tests for individual groups. $P$-values of these comparisons are $P$ = 0.06 Atp1a3 +12i, **$P$ < 0.01 Atp1a3+ 12i &LP, ****$P$ < 0.0001 Atp1a3 T705A + DMSO.

potential firing (Azarias et al, 2013). Therefore, to investigate whether T705 is regulated by neuronal activity, we treated rat primary cultures for 30 min with different compounds in order to stimulate or block neuronal activity. Neither increase in extracellular potassium (KCl)–mediated neuronal depolarization, nor glutamatergic receptor stimulation (glutamate), had an effect on relative pT705 intensity levels (Glu 92.5 ± 14.6; KCl 95.9 ± 13.4; Fig 4C). Ouabain did not change T705 phosphorylation levels (Oua 95.0 ± 19.5). Interestingly, we observed that blocking voltage-gated sodium channels with tetrodotoxin (TTX), which reduces baseline Na$^+$ influx in these spontaneously firing neurons, increased T705 phosphorylation (TTX 200 ± 17.1), while total levels of Atp1a3 remained unchanged (control 100 ± 9.63; Fig 4C). Ouabain-mediated inhibition of NKA is known to cause an increase in intracellular Na$^+$ concentration. Similarly, KCl treatment also results in an initial increase in intracellular Na$^+$, as introducing extracellular K$^+$ causes depolarization leading to the stimulation of voltage-gated Na$^+$ channels resulting in Na$^+$ influx. In contrast, TTX decreases internal Na$^+$ by blocking spontaneous bursts of action potentials, which high-density cortical cultures display (Mazzoni et al, 2007). Our results suggest that Atp1a3 T705 phosphorylation may be occurring under conditions where intracellular Na$^+$ is reduced.

**Atp1a3 phosphorylation alterations in trafficking**

We next investigated whether phosphorylation at this novel site could play a regulatory role in Na$^+$/K$^+$-ATPase trafficking. It is well known that ER assembly of NKA α- with β-subunits is required for export of the pump to the Golgi apparatus, with the β-subunit acting as a molecular chaperone to facilitate maturation and correct targeting of the pump to the plasma membrane (Geering, 2001, 2008). Prior to NKA insertion at the cell surface, α-subunits must be assembled with β-subunits to form a functioning enzyme. Therefore, we first evaluated whether phosphomutant α3-subunits were able to correctly assemble with β-subunits in order to reach the plasma membrane. Since expression of the β3 isoform is mainly in the lung, liver, and testis rather than in the brain (Malik et al, 1996; Arystarkhova & Sweadner, 1997), we examined only the ability of β1- and β2-subunits to interact with various Atp1a3 phosphomutant α-subunits. In addition to the Atp1a3 TA mutant, we generated a putative phosphomimetic Atp1a3 T705D (TD) mutant. We co-transfected HA-tagged Atp1a3 WT, TA, or TD phosphomutants with either YFP-β1 or YFP-β2 into HEK 293T cells and then immunoprecipitated α3-subunits with anti-HA antibody conjugated beads. We found that β subunits, detected using a GFP antibody,

precipitated with all of the Atp1a3 phosphomutants, indicating that they are capable of assembly with both YFP-β1 and YFP-β2 (Fig 5A).

However, the Atp1a3 phosphomutants immunoprecipitated different glycosylated forms of the β-subunit (Fig 5A). Structurally, the extracellular domain of the β-subunit has three N-linked glycosylation sites and three disulphide bonds conserved among all isoforms. We observed that there are two main N-glycosylated forms of the β-subunit, with a broader, higher molecular weight band (~80 kD β1, ~90 kD β2) representing mature subunits with complex-type glycans at the plasma membrane and a lower, sharper band (~65 kD β1, ~75 kD β2) representing intracellular immature subunits with only high mannose–type N-glycans in the ER (Tokhtaeva et al, 2010) (Fig 5A). We show that Atp1a3 WT precipitated both glycosylated forms of the β1- and β2-subunits, with a preference for the higher band indicative of mature subunits at the plasma membrane (Fig 5A). In contrast, the Atp1a3 TA phosphomutants did not heavily associate with the higher glycosylated form of the β2-subunit (YFP-β2/TA 16.3 ± 7.12; Fig 5A), and although the TA phosphomutants did associate with both glycosylated forms of the β1-subunit, the higher band showed a large reduction when normalized to WT (YFP-β1/TA 58.3 ± 8.98; Fig 5A), suggesting fewer heterodimers of this assembly at the plasma membrane. There was no change observed in the relative intensity of the lower intracellular band of TA mutants normalized to WT when co-transfected with YFP-β1 (101 ± 3.37), but there was a decrease found when co-transfected with YFP-β2 (82.7 ± 7.63; Fig 5A). The putative phosphomimetic TD only interacted with the lower molecular weight β subunit, indicating ER retention (YFP-β1/TD top band 6.59 ± 1.18; YFP-β2/TD top band 2.81 ± 1.15; YFP-β1/TD bottom band 95.0 ± 3.71; YFP-β2/TD bottom band 80.5 ± 7.97; Fig 5A). To further assess whether TA and TD mutants are able to reach the plasma membrane, surface biotinylation experiments were performed. β-Subunits with complex-type glycans seen at higher molecular weight were highly enriched in the surface fraction (Fig S4A), in agreement with previous studies. However, the lower molecular weight β-subunits also seem to be present at the surface (Fig S4). There was a large reduction in TA mutants at the surface (YFP-β1/TA 44.1 ± 3.9; YFP-β2/TA 57.7 ± 2.7; Fig S4A). As TD is likely sequestered in the ER, the TD bands observed (YFP-β1/TD 15 ± 1.4; YFP-β2/TD 46.1 ± 4.7; Fig S4A) suggest insufficient washing to remove residual biotin prior to lysis, which may indicate a further reduction in TA at the surface than observed. These results indicate that the phosphomimetic TD mutant does not leave the ER and surface expression of the non-phosphorylatable TA mutant is compromised, indicating a role for T705 in trafficking.

Next, we evaluated changes in the distribution of Atp1a3 upon pharmacological inhibition of the NAK kinase family in COS7 cells using light microscopy. Since Atp1a3 can be phosphorylated by AAK1 as well as GAK in cells (Fig 3C), we decided to use a GAK inhibitor 12i IVAP-1967 (Kovackova, Chang et al, 2015b) and a potent inhibitor for NAK family kinases AAK1 and BIKE, LP-935509 (Kostich et al, 2016). For estimating NAK distribution within cells, we used the FIJI plugin, Automated Detection and Analysis of ProTrusions (ADAPT) (Barry et al, 2015). ADAPT maps fluorescence distributions by successively eroding a binary representation of a cell and calculating the mean fluorescence intensity at the periphery of the binary cell at each iteration of the erosion operation. We observed that in COS7 cells

that are transfected with Atp1a3 with excess GFP-tagged β subunit, Atp1a3 localizes mostly to the cell periphery (Fig 5B and C). This preferential peripheral distribution is significantly reduced upon simultaneous treatment with NAK family kinase inhibitors 12i IVAP-1967 (12i) and LP-935509 (LP) (Fig 5B–D). Atp1a3 T705A does not preferentially localize to the periphery (Fig 5B–D), in agreement with reduced surface expression shown in surface biotinylation experiments. We compared these distributions by evaluating their peak intensity near the cell periphery, a means of estimating the extent of membrane localization, as the peripheral peak corresponds to the surface localization for membrane proteins such as the Na$^+$/K$^+$-ATPase (Fig 5D). Inhibition using only GAK inhibitor 12i IVAP-1967 had a smaller effect on Atp1a3 distribution, although not statistically significant (Fig 5D), while dual inhibition caused reduction in Atp1a3 localization to the cell periphery. This set of experiments support the notion that phosphorylation of Atp1a3 by NAK family kinases is necessary for its surface expression.

## Phosphomutant Atp1a3 accumulates in endosomal compartments in neurons

In order to examine localization of Atp1a3 in neurons, we co-transfected GFP-tagged Atp1a3 WT or TA with DsRed into rat hippocampal primary cultures. Atp1a3 WT showed typical surface distribution of Atp1a3 along the surface of dendrites and at dendritic spines, as previously reported (Blom et al, 2016) (Fig 6A). In contrast, the Atp1a3 TA phosphomutant formed clusters of large puncta in dendrites (Fig 6A). We repeated this experiment with HA-tagged Atp1a3 WT and TA co-transfected with GFP and found similar results (Fig S5A). Interestingly, HA-tagged TD showed diffuse distribution, but did not appear to localize to the surface of dendrites or to dendritic spines, consistent with ER retention (Fig S5A). To confirm that Atp1a3 TD is retained in the ER, we co-expressed HA-tagged Atp1a3 TD with either mCherry-tagged ER-3 (calreticulin) or mCherry-tagged Golgi-7 (β-1,4-galactosyltransferase 1) in rat hippocampal neurons. We observed that while there was a high amount of co-localization of TD with mCh-ER3 in the ER, there was no apparent co-localization with the mCh-Golgi-7 marker (Fig S5B), supporting its retention in ER. We did not see co-localization with endogenous LAMP1, even when lysosomal degradation is inhibited by a lysosomal inhibitor cocktail (200 μM chloroquine, 200 μM leupeptin, and 10 mM NH$_4$Cl) for 1 h, indicating that lack of phosphorylation is not targeting NKA to lysosomes (Fig S5C).

To determine at which step of the endocytic trafficking pathway Atp1a3 TA-containing pumps were being retained, we used endogenous stainings or co-transfections with different intracellular membrane compartment markers in rat hippocampal neurons expressing Atp1a3 WT and TA. We found that a subset of puncta in WT and in TA mutants co-localize with endogenous stainings for a sorting endosome marker, early endosome antigen 1 (EEA1) (Fig 6B, lower magnifications of these neurons are shown in Fig S6A and B). We quantified the number of Atp1a3 puncta along the length of dendrites, imaging dendrites with comparable thickness. The density of intracellular Atp1a3 accumulations in dendrites was significantly higher in TA (0.45 ± 0.10 puncta/μm) when compared to WT (0.11 ± 0.04 puncta/μm) (P = 0.017, n = 7 cells each). We observed substantial co-localization with Atp1a3 WT and Atp1a3 TA; however,

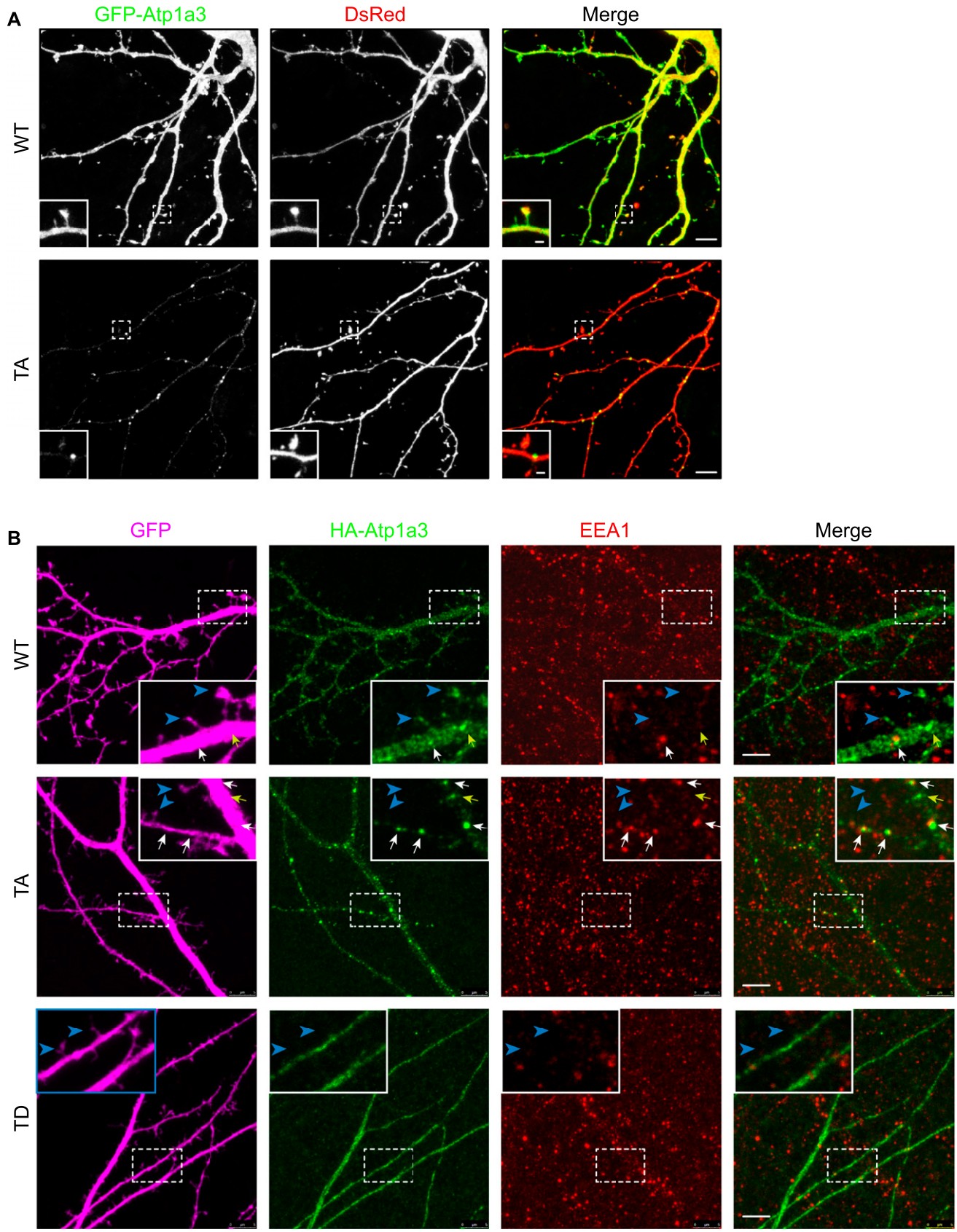

the percentage of puncta that co-localize with EEA1 was not significantly different (25 ± 14% for WT and 39 ± 8% for TA). In addition, we found that co-transfections of Atp1a3 WT or Atp1a3 TA with markers for early endosomes (mCh-Rab5), recycling endosomes (mCh-Transferrin receptor; mCh-TfR), and late endosomes (mCh-Rab7a) all showed some degree of co-localization, indicating that Atp1a3 trafficking is not fully dependent upon T705 phosphorylation (Fig S7A). When we quantified the percentage of Atp1a3 WT or TA puncta co-localization with these endosomal markers, there were no differences observed in mCh-Rab5 (in %: WT 21.7 ± 4.2 [$n$ = 3 cells]; TA 24.3 ± 8.3 [$n$ = 5 cells]; $P$ = 0.83), mCh-Rab7a (in %: WT 44.1 ± 6.4; TA 43.9 ± 10.5; $P$ = 0.98, $n$ = 5 cells, each), or mCh-TfR (in %: WT 50.8 ± 7.9; TA 31.3 ± 8.2; $P$ = 0.13, $n$ = 5 cells, each). Together with endogenous EEA1 stainings, these data indicate that similar percentage of Atp1a3 containing intracellular compartments co-localize with endocytic pathway markers; however, significantly more intracellular puncta are found in the TA mutant, specifically. Collectively, these data suggest that if Atp1a3 T705 cannot be phosphorylated, the NKA subsequently cannot be efficiently re-directed from sorting endosomal compartments to the plasma membrane, supporting its functional importance.

Since our data indicate a reduced surface expression of phosphomutant Atp1a3, we tested whether phosphomutants are functional by making use of ouabain (Oua), a cardiac glycoside known to inhibit the Na$^+$/K$^+$-ATPase. We generated ouabain-resistant (OuaR) mutants of Atp1a3 by introducing Q108R and N119D mutations (Price & Lingrel, 1988) to all of the Atp1a3 phosphomutants. OuaR WT, TA, or TD mutants were co-transfected with either YFP-$\beta$1 or YFP-$\beta$2 into COS7 cells and treated daily with 5 $\mu$M ouabain to observe survival and recovery. Only cells that have integrated exogenous cDNA encoding NKA ouabain resistance into their genome are able to survive ouabain selection (Clapcote et al, 2009). Within 48–72 h of ouabain treatment, COS7 cells transfected with OuaR WT and YFP-$\beta$1 or YFP-$\beta$2 began to assume normal growth, enough to be passaged. However, none of the other OuaR phosphomutants were able to recover, and by the end of one week of ouabain treatment, 80–90% of all cells were dead and past recovery consistently, from three independent experiments. These observations suggest that the phosphorylation state of Atp1a3 T705 is critical for its function.

## Na$^+$/K$^+$-ATPase function is affected in GAK NexCre conditional knockout mice

We next investigated the role of GAK in a knockout model, with a focus on putative substrate-linked phenotypes. We crossed Gak$^{tm2Legr}$ (Gak$^{flox}$) mutant mice with mouse lines expressing Cre recombinase in different neuronal subsets. We observed that conditional knockout of GAK in Drd1a-expressing striatal neurons (Tg(Drd1a-Cre))$^{EY262Gsat}$) or in midbrain dopaminergic neurons (Dat$^{IREScre}$) resulted in embryonic lethality. Curiously, conditional knockout of GAK in Neurod6$^{tm1(cre)Kan}$ (Nex$^{Cre}$) mice, specifically in the excitatory pyramidal neurons of the neocortex and hippocampus, resulted in viable and healthy offspring with no discernible behavioural or physical phenotypes.

Expression of GAK protein is reduced in Western blot analysis of neocortical lysates from mutant Nex$^{Cre/+}$/GAK$^{flox/flox}$ mice (F/F) compared to control Nex$^{Cre/+}$/GAK$^{flox/+}$ (F/+) (57.8 ± 2.70 and 100 ± 5.45, respectively; Fig 7A). Residual GAK expression can likely be attributed to glia or to non-excitatory neurons such as inhibitory neurons. Atp1a3 expression was increased in mutant mice (F/+ 100 ± 3.37; F/F 116 ± 4.28), but no change was observed in PSD-95 levels (F/+ 100 ± 3.94; F/F 89.5 ± 4.45; Fig 6A). We could not assess the levels of Atp1a3 pT705 as the antibody does not detect endogenous pT705 in mouse tissue.

To examine the electrophysiological properties of Nex$^{Cre}$/GAK mutant mice, we made whole-cell patch clamp recordings from CA1 pyramidal neurons using acute sagittal slices from P18-22 mice. The firing characteristics and intrinsic properties did not differ between control and mutant neurons (Fig S8A), indicating that loss of GAK is largely compensated by other kinases. Next, we directly assessed whether Na$^+$/K$^+$-ATPase function in GAK conditional knockout mice is partly affected. Since Atp1a3 function is critical for Na$^+$ clearance during action potential firing, neurons were injected with a positive holding current to have action potential firing during current-clamp recordings. Prior to ouabain application, there were no differences in input resistance, RMP, or the applied current between control F/+ or mutant F/F neurons (Table 2). Blocking NKA pump with 30 $\mu$M ouabain caused a rapid increase in firing rate and RMP, as expected. We observed a significantly larger RMP change in mutant F/F neurons than in F/+ controls (17.2 ± 2.2 mV, 14 cells from four animals and 9.5 ± 0.8 mV, 13 cells from five animals, respectively; Fig 7B), indicating that GAK mutants have compromised Na$^+$/K$^+$-ATPase function during firing. Similar observations of a large change in membrane potential upon ouabain application were reported in neurons from DJ-1 knockout mice, where mutations in the DJ-1 gene lead to an inherited form of early-onset parkinsonism (Pisani et al, 2006).

To further examine the functional consequence of the Atp1a3 T705 site, we generated a clustered regularly interspaced short palindromic repeats (CRISPR) mouse with T705 mutated to an alanine. Heterozygous mice were crossed to obtain homozygous Atp1a3 T705A phosphomutant mice. Out of 12 litters totalling 48 pups, there have been no homozygous offspring carrying this mutation thus far, indicating homozygous embryonic lethality consistent with a critical role of this site (Table 3).

# Discussion

In this study, we utilized a chemical genetics method with tandem mass spectrometry to identify a number of direct GAK substrates and their GAK phosphorylation sites. Among the substrates

**Figure 6. Atp1a3 phosphomutants show altered trafficking in neurons.**
**(A)** GFP-tagged Atp1a3 WT and phosphomutant TA co-transfection with DsRed in DIV15 rat hippocampal neurons demonstrate different localization patterns. Scale bars: 5 $\mu$m (dendrites), 1 $\mu$m (spines). **(B)** Neurons transfected with GFP (magenta) as cell fill and HA-ATP1a3 (HA immunostaining shown in green). Cultures are immunostained for endogenous EEA1 (red). Merge between HA-Atp1a3 and EEA1 is shown in the right panel. A zoomed-in image of a section of dendrite outlined with dashed lines is shown in insets. WT Atp1a3 is found dendritic spines (blue arrowheads), whereas Atp1a3 T705A is not found in spines (blue arrowheads). Atp1a3 T705A accumulates in intracellular compartments more frequently than WT Atp1a3, a subset of intracellular Atp1a3 puncta co-localizes with EEA1 (white arrows) and while others do not co-localize (yellow arrows). Atp1a3-T705D does not display any concentrated accumulations in dendrites and does not localize to spines (blue arrowheads). Scale bars = 5 $\mu$m.

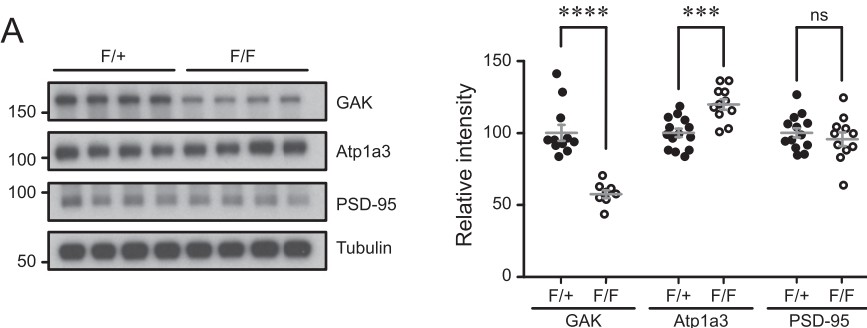

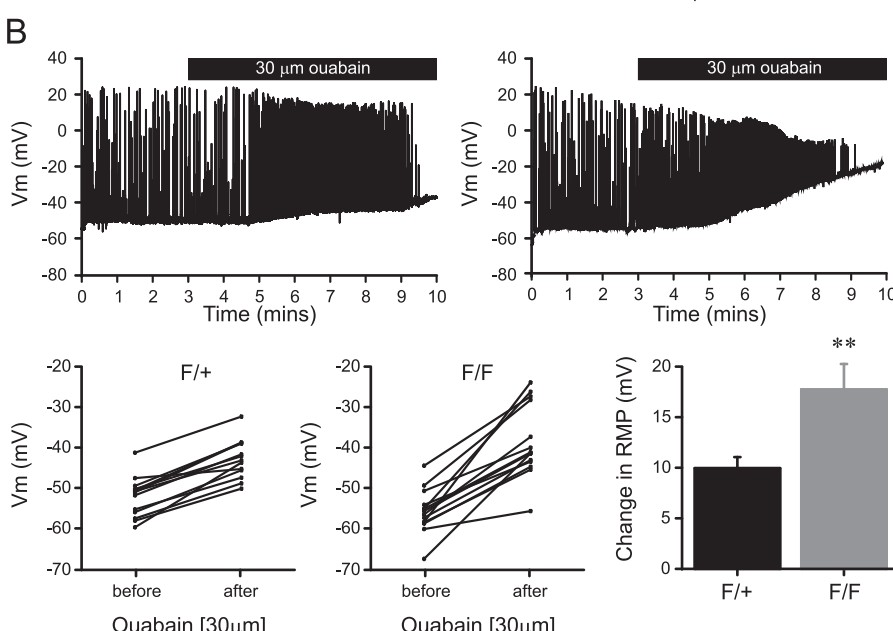

**Figure 7. Na⁺/K⁺-ATPase function is affected in GAK NexCre conditional knockout mice.**
**(A)** Cortical lysates of NexCre/GAK conditional knockout mice show a reduction in GAK levels ($P = 1.0 \times 10^{-5}$) and a slight increase in Atp1a3 levels ($P = 0.0003$) but not in PSD-95. $n = 11$ F/+ and 8 F/F animals. Graph data represent means ± SEM, $*P < 0.05$, $**P < 0.01$, $***P < 0.001$, $****P < 0.0001$, $t$ test. **(B)** Whole-cell patch clamp recordings from CA1 pyramidal neurons show a rapid increase in firing rate and RMP upon application of 30 μM ouabain. Mutant F/F neurons show a significantly larger change in RMP than F/+ controls. Graph data represent means ± SEM, $**P < 0.01$, $P = 0.003$.

identified, we validated the sodium potassium pump α3-subunit Atp1a3 as a GAK target at the novel site T705. We further show that the Atp1a3 T705 site is important in NKA trafficking and function and that its function is impaired in a GAK conditional knockout model.

GAK, also called DNAJ26, is a member of the J-domain–containing protein family also called heat-shock cognate 40 (HSP40s). J-domain proteins bind heat-shock 70 kD proteins (HSP70s) and act as co-chaperones during ubiquitous processes including protein folding and degradation. It is important to note that GAK is the only protein in mammals that contains both a kinase and a J-domain (Kampinga & Craig, 2010). An intriguing speculation is that one function of the GAK domain is to complement its J-domain roles by phosphorylating proteins targeted by HSP70, such as the Na⁺/K⁺-ATPase (Riordan et al, 2005) during trafficking of cargo. GAK's role in phosphorylating the μ-subunits of the adaptor proteins AP-1 and AP-2 is well characterized (Umeda et al, 2000; Korolchuk & Banting, 2002; Kametaka et al, 2007), along with its role in clathrin uncoating (Zhang et al, 2005b; Lee et al., 2005, 2006).

NAK family kinases are conserved from yeast to mammals. In yeast, loss-of-function mutations of the NAK family kinases prk1p and ark1p cause accumulations of actin-associated endocytic vesicles that cannot traffic further to other membrane compartments (Sekiya-Kawasaki et al, 2003). In mammals, AAK1 regulates endocytic trafficking (Conner & Schmid, 2002) and trafficking of cell surface receptors. AAK1 regulates notch receptor recycling and signalling (Gupta-Rossi et al, 2011), and neuregulin-1/Erbb4 signalling by altering Erbb4 trafficking (Kuai et al, 2011). In our loss-of-function analysis enabled by specific GAK and AAK1/BIKE inhibitors, we find that there is a significant reduction in NKA's peripheral distribution in COS7 cells. This finding is in agreement with reduced

**Table 2. Intrinsic property values of neurons prior to ouabain application.**

|  | F/+ | F/F | P-value |
| --- | --- | --- | --- |
| RMP (mV) | −51.5 ± 1.3 | −54.4 ± 1.4 | 0.14 |
| Input resistance (MΩ) | 203.6 ± 49.5 | 220.4 ± 52.8 | 0.75 |
| Current injection (pA) | 35.8 ± 4.4 | 37.7 ± 6.0 | 0.81 |

**Table 3. Atp1a3 T705A CRISPR mouse.**

| Litter | Wildtype +/+ | Heterozygous ATP1A3[T705/+] | Homozygous ATP1A3[T705/T705] |
|---|---|---|---|
| 1 | 2 | 2 | 0 |
| 2 | 3 | 3 | 0 |
| 3 | 2 | 3 | 0 |
| 4 | 1 | 3 | 0 |
| 5 | 1 | 1 | 0 |
| 6 | 1 | 2 | 0 |
| 7 | 2 | 0 | 0 |
| 8 | 1 | 0 | 0 |
| 9 | 3 | 3 | 0 |
| 10 | 5 | 3 | 0 |
| 11 | 2 | 3 | 0 |
| 12 | 1 | 1 | 0 |
| Total born | 24 | 24 | 0 |
| Percentage of total | 50 | 50 | 0 |

phosphomutant surface expression and increased presence of NKA in intracellular compartments. Therefore, it is possible that multiple members of the NAK family, including GAK and AAK1, could play complementary roles in receptor recycling via phosphorylating cargo proteins. Future experiments involving knockdown of GAK, AAK1, and possibly other NAK family members in neuronal cultures followed by evaluation of Atp1a3 trafficking would provide further support for their role in Atp1a3 trafficking.

In conjunction with known functions of NAK family kinases in membrane trafficking, we propose a simple model whereby GAK phosphorylates cargo proteins such as Atp1a3 and Sipa1L1 during or following endocytosis, with this phosphorylation being essential for efficient progression of Atp1a3 through subsequent trafficking steps, particularly for recycling back to the plasma membrane (Fig 8A). Previous work implicated PKA and PKC kinases in regulating NKA recycling in COS-1 cells (Kristensen et al, 2003). Our study demonstrates that without NAK family kinases or Atp1a3 T705 phosphorylation, NKA trafficking and function are impaired.

Importantly, due to the overlapping nature of substrate specificity among NAK family kinases, we cannot conclude that the identified putative substrates (Table 1) are phosphorylated by GAK and/or other NAK family kinases in organisms. In addition, all putative phosphorylation sites need to be validated by kinase assays.

### Regulation of sodium potassium pump trafficking by NAK family kinases

For proper functioning of the sodium potassium pump, the pump needs to be correctly distributed to the plasma membrane, a function that is regulated by kinases. NKA phosphorylation by PKA has been shown to increase cell surface expression of NKA in mammalian kidney collecting duct cells (Vinciguerra et al, 2003), while in human skeletal muscle cells, insulin activity recruits NKA to the surface via phosphorylation by ERK1/2 kinases (Al-Khalili et al,

2004). Here, we show that a novel site at Atp1a3 T705 is phosphorylated by GAK, and other NAK family kinases, and is critical for Atp1a3 localization and pump function. A putative phosphomimetic mutant, Atp1a3 T705D, demonstrated ER retention, suggesting that successful NKA exit from the ER is reliant on this site not being phosphorylated. However, a non-phosphorylatable Atp1a3 T705A mutant demonstrates that once the pump has left the ER, phosphorylation at this site is necessary for the pump to be sorted to the plasma membrane, as phosphomutant Atp1a3 T705A has reduced surface expression. Increased total Atp1a3 levels in GAK knockout mice indicate alterations in NKA function, possibly reflecting NKA levels increased in response to reduced functionality.

The NKA can be regulated by either a change in intracellular $Na^+$ concentration or a change in $Na^+$ affinity (Toustrup-Jensen et al, 2014). We see an increase in pT705 levels during TTX treatment. TTX blockade of voltage-gated $Na^+$ channels, while not changing RMP, would inhibit $Na^+$ influx, resulting in a lower intracellular $Na^+$ concentration. T705 phosphorylation may have a function in regulation of NKA in response to lower intracellular $Na^+$ concentrations and reduced firing, perhaps facilitating its intracellular trafficking.

### GAK in mouse models

In NexCre-mediated GAK conditional knockout mice, we observed no overall defects in either RMP or neuronal excitability. This contrasts with the embryonic or neonatal lethality observed when GAK is deleted in all neurons (Lee et al, 2008), dopaminergic neurons, or Drd1a-expressing medium spiny neurons in the striatum. This differential effect could be due to a compensatory mechanism present in pyramidal neurons, such as other NAK family kinases.

### GAK and ATP1A3 in neurodegenerative disease and aging

*GAK* has been identified as a gene associated with Parkinson's disease in several genome-wide association studies screens (Pankratz et al,

**A**

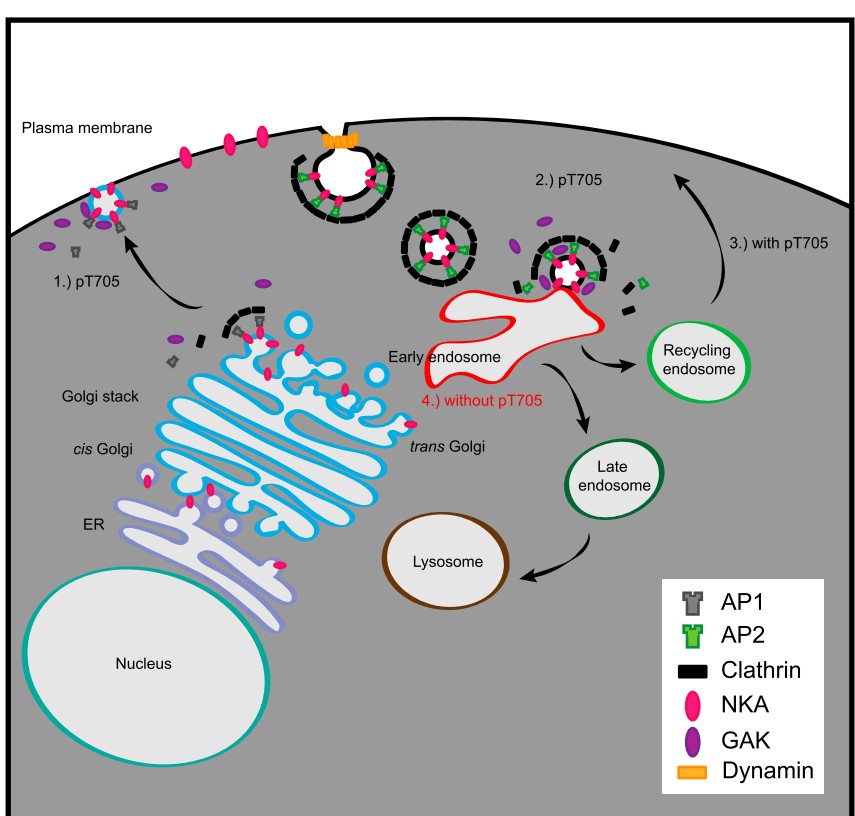

**Figure 8. GAK function in cargo trafficking.**
(A) A simplified model of GAK function during NKA cargo transport. (1) AP-1 and clathrin mediate transport from the Golgi to the plasma membrane during which Atp1a3 T705 may be phosphorylated by GAK. (2) AP-2 and clathrin mediate the endocytosis of NKA. During uncoating, phosphorylation of Atp1a3 T705 may occur. (3) Under basal conditions, GAK phosphorylates NKA at Atp1a3 T705, and NKA is recycled to plasma membrane. (4) Without T705 phosphorylation, NKA is preferentially localized to EEA1 containing early endosomes.

2009; Hamza et al, 2010; Rhodes et al, 2011; Pickrell et al, 2016). In addition, *AAK1* is associated with Parkinson's disease in a genome-wide association study (Latourelle, 2009). GAK is shown to interact with α-synuclein in a pathway involved with PD pathogenesis (Dumitriu et al, 2011) as well as to form part of a binding complex with another known PD-risk gene leucine-rich repeat kinase 2 (LRRK2) (Beilina et al, 2014). In a Drosophila model of PD, the GAK ortholog auxilin has been shown to underlie locomotor defects along with dopaminergic neuron loss (Song et al, 2017). There is also evidence that a number of PD-associated genes, including *LRRK2* and *GAK*, disrupt protein trafficking and degradation via the endosomal pathway as a consequence of age-related pathophysiology (Perrett et al, 2015).

Following neuronal activity, intracellular sodium clearance is mainly attributed to the α3-subunit (Azarias et al, 2013). Autosomal dominant mutations in *ATP1A3* have been linked to rapid-onset dystonia parkinsonism (de Carvalho Aguiar et al, 2004; Bottger et al, 2011), alternating hemiplegia of childhood, and CAPOS syndrome, with more than 80 different disease-associated mutations reported, many of which target ion binding sites (Clausen et al, 2017). In addition, studies into neurodegenerative disease mechanisms have shown *ATP1A3* deficits in a number of cases. In Alzheimer's disease (AD), there is reduced expression of Atp1a3 but not Atp1a1 in the frontal cortex of AD patients (Chauhan et al, 1997) as well as impaired NKA activity (Hattori et al, 1998). In AD, amyloid-β oligomers form unique assemblies called amylospheroids that target Atp1a3 and impair α3-containing NKA activity, leading to presynaptic calcium overload and neurodegeneration (Ohnishi et al, 2015). In PD, α-synuclein assemblies also target Atp1a3 by sequestering α3-containing NKA into clusters, ultimately affecting the sodium gradient (Shrivastava et al, 2015). These data support the possibility that deficiencies in GAK function in phosphorylating and regulating Atp1a3 may contribute to PD pathogenesis.

Our study uncovers novel roles for GAK in neurons, advancing our understanding of GAK function in cells. The conserved substrates and phosphorylation sites that we report will enable exploration of related functions in other tissues. In addition, the phosphorylation sites can be used to measure GAK activity in different tissues, during development and in PD disease models.

# Materials and Methods

### Mouse maintenance

Breeding and experiments were performed under the Animals (Scientific Procedures) Act 1986 of the United Kingdom and approved by institutional ethical review. Mice were group-housed and maintained on a 12-h light/dark cycle, with food and water provided ad libitum. Each mouse strain was backcrossed into C57 Bl/6 (Jackson) for at least three generations. None of the experimental mice were immunocompromised. Both male and female mice

were used and randomly allocated to experimental groups. Developmental ages between embryonic day (E)16 and postnatal day (P)30 were used as indicated in the figure legends.

GAK (B6;129S6-Gak[tm2Legr], MMRRC stock JAX: 36793) (Lee et al, 2008), Dat[IREScre] (B6.SJL-Slc6a3[tm1.1(cre)Bkmn/J], IMSR Cat. No. JAX:006660, RRID: IMSR_JAX:006660) (Backman et al, 2006), and Thy1-YFP ((Thy1-YFP)[HJrs], IMSR Cat. No. JAX:003782, RRID:IMSR_JAX:003782) (Feng et al, 2000) were obtained from Jackson Laboratories. Nex[Cre] (Neurod6[tm1(cre)Kan], MGI Cat. No. 4429523, RRID:MGI:4429523) was obtained from Marcus Schwab (Schwab et al, 2000) and Drd1aCre (Tg(Drd1a-cre)[EY262Gsat], MGI:3836631) was obtained from MMMRC UC Davis (Gong et al, 2007).

Atp1a3 T705A was created in house using the CRISPR system.

For experiments, NexCre homozygous GAK heterozygous males (Nex[Cre/Cre];GAK[flox/+]) were bred with GAK homozygous (GAK[flox/flox]) females to generate experimental controls Nex[Cre/+];GAK[flox/+] (F/+) or mutants Nex[Cre/+] GAK[flox/flox] (F/F). All mice were bred and genotyped as recommended by Jackson Laboratories.

### Isolation of primary cells

For rat primary cultures, E18 embryos of either sex from Sprague–Dawley rats (Charles River Labs) were used. After removal of the meninges, cortices and hippocampi were dissected, separately pooled, and mechanically triturated. Dissociated neurons were then plated in plating medium containing 10% foetal bovine serum (Hyclone), 0.45% dextrose, 0.11 mg/ml sodium pyruvate, 2 mM glutamine, 100 units/ml penicillin, and 100 mg/ml streptomycin in modified Eagle's medium. Hippocampal neurons were seeded onto 18-mm glass coverslips coated with 0.06 mg/ml poly-D-lysine (Sigma-Aldrich) and 0.0025 mg/ml laminin (Sigma-Aldrich) at a density of approximately $0.15 \times 10^6$ cells per coverslip. Cortical neurons were seeded onto either 6-well or 12-well plates coated with poly-D-lysine/laminin at a density of $0.3 \times 10^6$ cells per well. After 4 hours, plating medium was replaced with maintenance media containing 1× B27 (Invitrogen), 100 units/ml penicillin and 100 mg/ml streptomycin, 2 mM glutamine and 12.5 $\mu$M glutamate in Neurobasal Media (Invitrogen). Half of the media was replaced with fresh about every four days to maintain cultures until use. Plasmid transfections in cultured neurons were performed with Lipofectamine-2000 (Invitrogen) according to manufacturer's instructions. For 12-well plates, 1 $\mu$g total plasmid DNA was transfected per well. For six-well plates, 3 $\mu$g of total plasmid DNA was transfected.

### Tissue culture cells

HEK 293T cells and COS7 cells were maintained in DMEM (Gibco) supplemented with 10% FBS and 1% penicillin–streptomycin. Transfections were performed using Xtremegene 9 (Roche) according to manufacturer's instructions. Cells were collected for Western blot analysis 48 h post- transfection.

### Plasmids and cloning

GAK mouse (Open Biosystems) FL and kinase domain–only (K) cDNA were cloned into a pRK5 mammalian expression vector containing an N-terminal hemagglutinin (HA) tag (HA-pRK5) by PCR cloning using 5′ SalI and 3′ NotI restriction sites and the following oligos:

GAK SalI Fwd: CATCGTGTCGACTATGTCGCTGCTGCAGTCTGC
GAK NotI Rev: TCGAGTGCGGCCGCTCAGAAGAGGGGCCTCGAGC
GAK[1–400] NotI Rev: ACAGTTGCGGCCGCCTAGTTAGCCACAGACTGGATGA

All GAK mutants were generated by site-directed mutagenesis using QuickChange (Agilent Technologies) and the following oligos:

GAK K69R (KD) Fwd: GGCAGAGAGTATGCATTAAGGCGATTACTATCC
GAK K69R (KD) Rev: GGATAGTAATCGCCTTAATGCATACTCTCTGCC
GAK T123A (AS) Fwd: GAGTTCCTCCTGCTTGCGGAGCTTTGTAAAGG
GAK T123A (AS) Rev: CCTTTACAAAGCTCCGCAAGCAGGAGGAACTC
GAK L121I/T123A (AS rescue) Fwd: GGGCAGGCTGAGTTCCTCATTCTTGCGGAGCTTTG
GAK L121I/T123A (AS rescue) Rev: CAAAGCTCCGCAAGAATGAGGAACTCAGCCTGCCC
GAK C87Q/F88I/L89M (AS rescue) Fwd: CAGGAAGTTCAGATCATGAAAAAACTTTCTGGCCAC
GAK C87Q/F88I/L89M (AS rescue) Rev: GTGGCCAGAAAGTTTTTTCATGATCTGAACTTCCTG

Atp1a3 mouse cDNA (Source Bioscience) and Sipa1L1 mouse cDNA (Source Bioscience) were cloned into HA-pRK5 by PCR cloning using 5′ SalI and 3′ NotI restriction sites (already present in Atp1a3) and the following oligos for Sipa1L1:

Sipa1L1 SalI Fwd: GGTGGTGGTGGTGGGTCGACCATGACCAGTTTGAAGCGGTCG
Sipa1L1 NotI Rev: TGCTGCTGCTGCGGCCGCCTAGCTCATGTCTATGG
Sipa1L1 Act1 NotI Rev: TGCTGCTGCTGCGGCCGCTCACTCCTCCTCCGTGCTCTG

All phosphomutants of Atp1a3 and Sipa1L1 were generated by site-directed mutagenesis using QuickChange and the following oligos:

Atp1a3 T705A Fwd: GCAATTGTGGCTGTGGCTGGCGATGGTGTGAATGAC
Atp1a3 T705A Rev: GTCATTCACACCATCGCCAGCCACAGCCACAATTGC
Atp1a3 T705D Fwd: GCAATTGTGGCTGTGGATGGCGATGGTGTGAATGAC
Atp1a3 T705D Rev: GTCATTCACACCATCGCCATCCACAGCCACAATTGC
Atp1a3 Q108R (ouabain resistance) Fwd: CTGGCCTATGGCATCCGGGCAGGGACGGAGGATGAC
Atp1a3 Q108R (ouabain resistance) Rev: GTCATCCTCCGTCCCTGCCCGGATGCCATAGGCCAG
Atp1a3 N119D (ouabain resistance) Fwd: GATGACCCTTCCGGTGACGACCTGTACCTGGGCATAGTG
Atp1a3 N119D (ouabain resistance) Rev: CACTATGCCCAGGTACAGGTCGTCACCGGAAGGGTCATC
Sipa1L1 T249A Fwd: CTCAGTGACTTCCTCATCGCTGGTGGGGGCAAGGGTTCTGG
Sipa1L1 T249A Rev: CCAGAACCCTTGCCCCCACCAGCGATGAGGAAGTCACTGAG

The mCh-ER3 (plasmid #55041; Addgene) and mCh-Golgi7 (plasmid #55052; Addgene) plasmids were gifts from Michael Davidson. The mCh-Rab5 (plasmid #49201; Addgene) and mCh-Rab7A

(plasmid #61804; Addgene) plasmids were gifts from Gia Voeltz (Friedman et al, 2010; Rowland et al, 2014). Lamp1-RFP (plasmid #1817; Addgene) was a gift from Walther Mothes (Sherer et al, 2003). The pEYFP-Na,K-ATPase-β1 (rat) and -β2 (human) subunit plasmids were a kind gift from Dr. Olga Vagin (UCLA). The mCh-Transferrin plasmid was a kind gift from Dr. Michael Ehlers (Duke University).

## Immunostaining

Cultured hippocampal neurons on glass coverslips were fixed for 10 min at RT with 4% paraformaldehyde/sucrose in PBS, permeabilized for 10 min RT with 0.1% Triton-X100 in PBS, and then blocked for 1 h RT with 10% normal goat serum in PBS. Cultures were incubated with primary antibodies in blocking buffer at 4°C overnight. After three washes in PBS, cultures were labelled with fluorescent secondary antibodies (1:500) for 1 h RT and washed three times with PBS and the coverslips mounted onto slides using Fluoromount-G (Southern Biotech). Antibodies used for immunocytochemistry are rat anti-HA (1:500; Roche), mouse anti-EEA1 (clone 14 1: 250; BD Transduction Laboratories), rabbit anti-LAMP1 (ab24170, 1:500; Abcam), and mouse anti-c-Myc (1:500; Thermo Fisher Scientific). Fluorescent secondary antibodies with minimal cross-reactivity to other species were obtained from Invitrogen or Jackson ImmunoResearch.

## Western blotting

For drug treatment experiments, DIV14-15 cultured cortical neurons in a 12-well plate were used. Each well was treated for 1 h in culture medium. Neurons were directly lysed with 1× sample buffer (4× stock; Invitrogen) containing 400 mM DTT. Lysates were denatured for 10 min at 95°C.

For CIP dephosphorylation experiments, DIV15 rat cortical cultures in 60-mm dishes were treated for 1 H with either DMSO, 0.5 $\mu$M OA, or 10 $\mu$M cyclosporin A and lysed in lysis buffer (20 mM Tris, pH 8.0, 150 mM NaCl, 1% NP40 [Igepal], 10% glycerol, 1× protease inhibitor cocktail [Roche]). Lysates were left to solubilize on ice for 10 min and then clarified by centrifugation for 10 min at 20,000 $g$. For controls, 10 $\mu$l of supernatant was removed and incubated with 35 $\mu$l H$_2$O and 5 $\mu$l 10× CutSmart buffer (NEB) for 1 H at 37°C. For CIP-treated samples, 10 $\mu$l of supernatant was incubated with 35 $\mu$l H$_2$O, 5 $\mu$l 10× CutSmart buffer, and 2 $\mu$l (20 units) of calf intestinal alkaline phosphatase (NEB) for 1 H at 37°C. To stop the reaction, 16 $\mu$l of 4× sample buffer was added and samples were denatured for 10 min at 95°C.

For rat embryonic and adult whole-brain lysates, tissues were solubilized in either 200 $\mu$l or 2 ml of 1× sample buffer, respectively, and sonicated. Lysates were then clarified by centrifugation for 10 min at 20,000 $g$. The resulting supernatant was further diluted in 1× sample buffer 1:16 or 1:12 for embryonic or adult lysates, respectively, and 5 $\mu$l was loaded.

For immunoprecipitation (IP) experiments in HEK 293T cells, cells were lysed in IP buffer (20 mM Tris, pH 8.0, 150 mM NaCl, 1% NP40 [Igepal], 10% glycerol, 1× protease inhibitor cocktail [Roche] and 1× phosphatase inhibitor cocktail 3 [Sigma-Aldrich]). Lysates were

solubilized for 30 min at 4°C and then clarified by centrifugation for 10 min at 20,000 $g$. An aliquot of input was reserved as needed, and the remaining supernatant was incubated with 30–40 $\mu$l of HA beads (clone 3F10; Roche) for 2 h rotating at 4°C. The beads were washed three times in IP buffer before proteins were eluted in 2× sample buffer and denatured for 10 min at 95°C.

Protein samples were subjected to SDS–PAGE separation on 4–12% gradient gels according to manufacturer's protocol (Invitrogen). Resolved proteins were transferred to polyvinylidene fluoride membrane, blocked in either 5% non-fat dry milk (Sigma-Aldrich) or 5% BSA (Sigma-Aldrich) for 1 h RT, and incubated overnight at 4°C with primary antibodies. Signal was detected using horseradish peroxidase (HRP)–conjugated secondary antibodies (Jackson) followed by a chemiluminescence reaction using Amersham ECL substrate (GE Healthcare). Chemiluminescent signal was detected using X-ray film (GE Healthcare). Densitometry was done using ImageJ, and values were analysed using $t$ test.

Antibodies used in immunoblots include rat anti-HA (3F10 1: 5,000; Roche), mouse anti-$\alpha$ tubulin (1:20,000; Molecular Probes), and mouse anti-Atp1a3 (1:30,000; Thermo Fisher Scientific). GAK rabbit antibody was a gift from Lois Greene used at 1:10,000. Phosphoantibodies were generated by Covalab. Atp1a3 phospho-T705 antibody was generated against the mouse Atp1a3 epitope TQRAGHRRIL-phospho-S-DV (1:500 for Western blot). Horseradish peroxidase–conjugated secondary antibodies against mice, rats, and rabbits were used between 1:10,000 and 1:25,000 (Jackson ImmunoResearch).

## Surface biotinylation

Cells were washed 2× with cold PBS and then incubated in 1× PBS containing 1 mg/ml sulpho-NHS biotin conjugate (Pierce) for 30 min rocking at 4°C. Cells were washed 2× to remove unbound biotin, then lysed in NP40 lysis buffer (20 mM Tris, pH 8.0, 150 mM NaCl, 1% NP40 [Igepal], 10% glycerol, 1× protease inhibitor cocktail [Roche], and 1× phosphatase inhibitor cocktail 3 [Sigma-Aldrich]), and centrifuged at 20,000 $g$ for 10 min. After reserving an aliquot for input, the remaining supernatant was incubated with 30 $\mu$l of streptavidin beads (Pierce) for 1 h rotating at 4°C. The beads were washed 3× in lysis buffer, and surface proteins were eluted in 2× sample buffer for 10 min at 95°C.

## Protein purification

GAK K constructs for protein purification, analog-specific (AS) and KD, were cloned into the pFastBac HT B vector (Invitrogen) by PCR cloning to contain N-terminal 6× His tag. His-tagged GAK-K-AS and GAK-K-KD baculovirus expression (Bac-to-Bac Baculovirus expression system; Invitrogen) was performed according to manufacturer's protocol in insect cells.

## Kinase assays

GAK assays: GAK was expressed in HEK 293T cells for 48 h maintained in 10% FBS, 5% pen/strep in Dulbecco's modified Eagle medium. Cells were lysed with lysis buffer containing 1% Nonidet

P-40 (Igepal), 10% glycerol, 1 mM $Na_3VO_4$, 20 mM $\beta$-glycerol phosphate, 50 mM NaF, 1× complete protease inhibitor cocktail (Roche), 1× phosphatase inhibitor cocktail 3 (Sigma-Aldrich) in 20 mM Tris–HCl pH 8.0 and 150 mM NaCl. Lysates were incubated on ice for 30 min and centrifuged at 20,000 $g$ for 15 min. Supernatant was precleared with IgG-Sepharose (GE Healthcare) for 30 min and incubated with Anti-HA Affinity matrix (clone 3F10; Roche) for 2 h at 4°C to immunoprecipitate HA-tagged GAK. Beads were washed twice with lysis buffer, once with lysis buffer containing 1 M NaCl for 10 min and once with lysis buffer for 10 min. After three additional washes with kinase reaction buffer (20 mM Tris–HCl pH 7.5, 10 mM $MgCl_2$, 1 mM DTT, 1 $\mu$M cyclic AMP–dependent protein kinase inhibitor peptide and 1 $\mu$M OA), beads were incubated in a kinase reaction mixture for 30 min at 30°C. Reaction volume was 30 $\mu$l in addition to the bead volume. Either 0.5 mM ATP-$\gamma$-S (Sigma-Aldrich) or 0.5 mM of the analog ATP-$\gamma$-S (6-Bn-ATP-$\gamma$-S, 6-PhEt-ATP-$\gamma$-S or 6-Furfuryl-ATP-$\gamma$-S from BioLog Life Science Institute) was included in the reaction. The reaction was immediately followed by a 1 h alkylation reaction at RT by adding 1.5 $\mu$l of 100 mM p-nitro mesylate (PNBM) per 30 $\mu$l of kinase reaction. The reaction was stopped by the addition of 4× sample buffer containing 400 mM DTT to a final concentration of 1×, and proteins on the beads were denatured at 95°C for 10 min. Supernatants were run on a Western blot. Thiophosphorylation was detected by anti-thiophosphate ester antibody 1:30,000 (Abcam).

## Chemical genetics for kinase substrate identification

### Substrate labelling

Mice were euthanized with cervical dislocation. Brains were washed in cold PBS and immediately transferred to ice cold lysis buffer (10 mM $MgCl_2$, 100 mM NaCl, 20 mM Tris pH 7.5, 0.5 mM DTT, 1× protease inhibitor cocktail [Roche], and 0.25% Nonidet P-40). Brains were minced and lysed by sonication, and lysates were centrifuged at 20,000 $g$ for 15 min. The protein concentration of the supernatant was measured by BCA assay (Pierce) and diluted to 10–20 $\mu$g/$\mu$l in the same lysis buffer. To 100 $\mu$l of supernatant, ~10 $\mu$g of purified GAK (either His tag purified from SF21 cells or freshly immunoprecipitated from HEK 293T cells on HA beads [30 $\mu$l resin volume]), 10 $\mu$M protein kinase C inhibitor (Bisindolylmaleimide I–Calbiochem), 1 $\mu$M cyclic AMP–dependent protein kinase inhibitor peptide, 3 mM GTP, 100 $\mu$M ATP, and 0.5 mM Furfuryl-ATP-$\gamma$-S were added for labelling reaction. PKC inhibitor and GTP were used to reduce background. Substrates were labelled for 1–2 h at 30°C on a nutator.

### Covalent capture

Covalent capture of thiophosphorylated substrate proteins was performed as described (Hertz et al, 2010) except for the following modifications. The labelled brain lysates were denatured by adding 60% by volume solid urea, 1 M TCEP to 10 mM and incubating at 55°C for 1 h. Proteins were then digested by diluting the urea to 2 M by addition of 100 mM $NH_4HCO_3$ (pH 8), adding additional TCEP to 10 mM final, 0.5 M EDTA to 1 mM, and trypsin (Promega) 1:20 by weight. The labelled lysates were digested for 16 h at 37°C, acidified to 0.5% TFA, and desalted using a sep pak C18 column (Waters)

eluting into 1 ml 50% acetonitrile 0.1% TFA. The desalted peptides were dried using a speed vacuum to 40 $\mu$l. The pH of the peptides was adjusted by adding 40 $\mu$l of 200 mM Hepes pH 7.0 and 75 $\mu$l acetonitrile and brought to pH 7.0 by addition of 10% NaOH. The peptide solution was then added to 100 $\mu$l iodoacetyl beads (Pierce) equilibrated with 200 mM Hepes pH 7.0 and incubated with end-over-end rotation at RT in the dark for 16 h. The beads were then added to small disposable columns, washed with $H_2O$, 5 M NaCl, 50% acetonitrile, 5% formic acid, and 10 mM DTT, followed by elution with 100 $\mu$l and 200 $\mu$l (300 $\mu$l total) 1 mg/ml oxone (Sigma-Aldrich), desalted, and concentrated on a 10-$\mu$l zip tip (Millipore) eluting into 60 $\mu$l total volume. The resulting phosphopeptide mixtures were resuspended in 35 $\mu$l 0.1% trifluoroacetic acid and injected three times (10 $\mu$l per injection), with each run a 1-h gradient elution with one activation method per run (collision-induced dissociation, multistage activation, and higher energy collisional activation dissociation). A LTQ-Orbitrap Velos was used for data acquisition. Data processing was performed using MaxQuant bioinformatics suite.

## Measurement of Atp1a3 distribution in COS7 cells using ADAPT ImageJ software plugin

COS7 cells were split at confluency and GFP-tagged $\beta$1 was co-transfected with either HA-tagged Atp1a3 WT or T705A. After 48 h, cells were treated with either DMSO or 2 $\mu$M LP for 1.5 h. Cell media was then replaced for 30 mins with a serum-free media including either DMSO or LP. This was followed by the addition of 5 $\mu$M 12i or DMSO into wells for another 2 h. DMSO control, 12i, LP or 12i+LP conditions all received the same total volumes of DMSO solvent. LP treatment was done for 4 h total and 12i treatment for 2 h total (in serum-free conditions). Cells were fixed with 4% PFA/sucrose for 10 min at RT and stained using anti-HA antibody. $\beta$1 expression was visually confirmed in all imaged cells.

## In vivo tissue processing

Nex[Cre/+];Gak[flox/flox];Thy1-YFP and Nex[Cre/+];Gak[flox/+];Thy1-YFP mice (P18-P20) were anaesthetized by intraperitoneal injection of 70–100 mg/kg ketamine + 10–20 mg/kg xylazine and perfused with ice cold PBS followed by 4% PFA. Brains were post-fixed in 4% PFA overnight at 4°C. Using vibratome sectioning, 50–100-$\mu$m-thick coronal sections were collected. Sections were mounted to glass slides using Fluoromount-G and imaged.

## Electrophysiology

Nex[Cre/+];Gak[flox/flox] and Nex[Cre/+];Gak[flox/+] mice (P18-P22) were anaesthetized by intraperitoneal injection of ketamine (80 mg/kg) and xylazine (10 mg/kg). Mice were killed by cervical dislocation. Brains were removed, the midline was cut down, and the cut surface was glued to the stage of the slicing chamber containing ice cold (~1°C) sucrose artificial cerebral spinal fluid (aCSF; composition in mM: 189 sucrose, 26 $NaHCO_3$, 2.5 KCl, 5 $MgCl_2$, 0.1 $CaCl_2$, 1.2 $NaH_2PO_4$, 10 glucose), gassed with 95% $O_2$–5% $CO_2$. Sagittal slices (300 $\mu$m thick)

were prepared and transferred into an interface storage chamber containing aCSF (composition in mM: 125 NaCl, 3 KCl, 1 MgCl$_2$, 2 CaCl$_2$, 1.2 NaH$_2$PO$_4$, 26 NaHCO$_3$, 10 glucose) bubbled with 95% O$_2$–5% CO$_2$ at RT. Slices were left in the interface storage chamber for at least 1 h before use and were used up to 8 h after slicing.

Whole-cell patch clamp recordings were made from the somata of hippocampal CA1 pyramidal neurons using infrared differential interference contrast optics. Slices were constantly perfused with 95% O$_2$–5% CO$_2$ oxygenated aCSF at a flow rate of ~ 3 ml·min$^{-1}$ at 31–32°C. Patch electrodes (3–7 MΩ) were pulled from borosilicate glass. Membrane currents were recorded using a Multiclamp 700B amplifier (Molecular Devices), sampled at 5 kHz, and low pass Bessel–filtered at 1 kHz. Compensations for slow and fast capacitive currents were performed. Data acquisition was controlled using Clampex 10.3.

Miniature excitatory postsynaptic currents were recorded at –75 mV using an intracellular solution comprising the following (mM): 140 CsCH$_3$SO$_4$, 8 NaCl, 10 Hepes, 0.5 EGTA, 2 Mg-ATP, 0.3 Na-GTP, 5 QX-314, and 0.2% biocytin; osmolarity ~285 mosmol·l$^{-1}$; pH 7.3 with CsOH. Miniature EPSCs were isolated with 10 $\mu$M gabazine and 1 $\mu$M tetrodotoxin (Tocris Chemicals). For firing studies, the intracellular solution comprised (mM) 120 K+-gluconate, 24 KCl, 4 NaCl, 4 MgCl2, 0.16 EGTA, 10 Hepes, 4 K2-ATP, and 0.2% biocytin, pH 7.2 adjusted with KOH.

### Electrophysiological data analysis

Cells were rejected if input/access resistance or holding current changed by more than 20% of the initial value after establishing the whole-cell configuration. For miniature excitatory postsynaptic currents, 5 min of data was analysed from each viable cell, and postsynaptic currents were manually detected using MiniAnalysis software (v6.0.3, Synaptosoft). The amplitude and area detection threshold was set to five times greater than the root mean square of the baseline noise. For firing studies, data were analysed with pClamp10 software (Molecular Devices). Statistical analysis was performed using Origin 2017 software (OriginLab). Data were tested for normality using a Shapiro–Wilk test; significant criterion was $\alpha$ = 0.05. Mann–Whitney $U$ test was used on non-normally distributed data.

# Supplementary Information

# Acknowledgments

We thank members of the Ultanir Lab for their feedback and thoughtful suggestions, Richard Li for his expertise in protein alignment and PYMOL modelling, Ian Rosewell and Sunita Varsani-Brown for their generation of the clustered regularly interspaced short palindromic repeats Atp1a3T705A mouse, David Barry for help with ADAPT ImageJ plugin and light microscopy analysis, and Lucas Baltussen for manuscript preparation. We thank Dr Hjalmar Brismar for GFP-tagged Atp1a3 plasmid. This work was supported by the Francis Crick Institute which receives its core funding from Cancer Research UK (FC001201), the UK Medical Research Council (FC001201), and the Wellcome Trust (FC001201).

## Author Contributions

AW Lin: conceptualization, data curation, formal analysis, investigation, methodology, project administration, and writing—original draft, review, and editing.
KK Gill: conceptualization, data curation, formal analysis, investigation, methodology, and writing—original draft, review, and editing.
MS Castañeda: data curation.
I Matucci: data curation.
N Eder: data curation and formal analysis.
S Claxton: data curation.
H Flynn: data curation and formal analysis.
AP Snijders: data curation, formal analysis, and supervision.
R George: resources, data curation, and methodology.
SK Ultanir: conceptualization, data curation, formal analysis, supervision, funding acquisition, investigation, methodology, project administration, and writing—original draft, review, and editing.

## Conflict of Interest Statement

The authors declare that they have no conflict of interest.

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
