## [Reviewer comments · Life Science Alliance]

Life Science Alliance

Chemical Genetic Identification of GAK Substrates Reveals its Role in Regulating Na⁺/K⁺-ATPase

Amy Lin, Kalbinder Gill, Noreen Eder, Marisol Sampedro Castañeda, Suzanne Claxton, Irene Matucci, Helen Flynn, Ambrosius P. Snijders, Roger George, and Sila Ultanir

DOI: [10.26508/lsa.201800118](https://doi.org/10.26508/lsa.201800118)

Corresponding author(s): Sila Ultanir, The Francis Crick Institute

Review Timeline:	Submission Date:	2018-06-26
	Editorial Decision:	2018-07-17
	Revision Received:	2018-10-29
	Editorial Decision:	2018-11-15
	Revision Received:	2018-12-07
	Accepted:	2018-12-10

Scientific Editor: Andrea Leibfried

Transaction Report:

July 17, 2018

Re: Life Science Alliance manuscript #LSA-2018-00118-T

Dr. Sila Ultanir
The Francis Crick Institute
1 Midland Road
London, London NW1 1AT
United Kingdom

Dear Dr. Ultanir,

Thank you for submitting your manuscript entitled "Chemical Genetic Identification of GAK Substrates Reveals its Role in Regulating Na⁺/K⁺-ATPase" to Life Science Alliance. The manuscript was assessed by expert reviewers, whose comments are appended to this letter.

As you will see, the reviewers provide constructive input on how to strengthen your work to warrant publication here. Most importantly, a proof of principle to show that GAK affects the phosphorylation and trafficking of endogenous Na⁺/K⁺-ATPase (Atp1a3) is currently missing, and such proof would be required for publication in Life Science Alliance. The reviewers note other issues that I am not listing individually here as they all seem straightforward to address. We would like to invite you to submit a revised version of your manuscript, addressing all points raised by the reviewers and particularly providing the above mentioned support for your conclusion on GAK-regulated Atp1a3 phosphorylation and trafficking.

- A letter addressing the reviewers' comments point by point.
- An editable version of the final text (.DOC or .DOCX) is needed for copyediting (no PDFs).
- High-resolution figure, supplementary figure and video files uploaded as individual files: See our detailed guidelines for preparing your production-ready images, <http://life-science-alliance.org/authorguide>
- Summary blurb (enter in submission system): A short text summarizing in a single sentence the study (max. 200 characters including spaces). This text is used in conjunction with the titles of

papers, hence should be informative and complementary to the title and running title. It should describe the context and significance of the findings for a general readership; it should be written in the present tense and refer to the work in the third person. Author names should not be mentioned.

B. MANUSCRIPT ORGANIZATION AND FORMATTING:

Full guidelines are available on our Instructions for Authors page, <http://life-science-alliance.org/authorguide>

Thank you for this interesting contribution to Life Science Alliance. We are looking forward to receiving your revised manuscript.

Sincerely,

Reviewer #1 (Comments to the Authors (Required)):

In the present study, Lin et al. employ extensive mutagenesis of the GAK kinase to enable a chemical screening approach to identify direct GAK substrates. While their screen fails to identify

some known substrates, it proved useful in identifying threonine 705 of the Potassium-Sodium exchanger (NKA) subunit ATP1a3 as a potential substrate. Through the use of phosphorylation resistant and mimetic mutants, the authors demonstrate that the phosphorylation of T705 is required for plasma membrane delivery and overall functionality of NKA. While most of the data is convincing, some important evidence is missing in the manuscript: The authors need to show that loss of GAK in a cellular system perturbs ATP1a3 trafficking and thus NKA function. They provide some indirect evidence from conditional knockout mice but I couldn't find direct evidence that loss of GAK results in a loss of ATP1a3 phosphorylation, trafficking and ultimately NKA function. I suggest to use their COS-7-Ouabain lethality model to test whether GAK knockdown (or knockout with Crispr/Cas9, or inhibitor treatment) also compromises NKA trafficking and function. As it is, most of the functional evidence stems from the phosphorylation mutant proteins, which leaves this reviewer slightly skeptical. Besides this criticism, I am overall supportive of publication as the manuscript is very extensive, provides a significant advance and mostly convincing evidence. Some additional concerns are listed under major and minor points.

Major points:

The authors show that the phosphorylation resistant TA mutant of Atp1a3 is being mis-sorted into lysosomes. It would be good to confirm this with an antibody against endogenous LAMP1 (or LAMP2 or CD63). At least in our hands, Lamp1-GFP is not a good marker when overexpressed as it is also subject to trafficking to and from lysosomes.

The authors use a heavily truncated fragment of GAK-AS for their substrate screen. Could this distort the substrate specificity? It is somewhat concerning that the known substrates AP-2 and PP2A, which should be present in brain lysates, was not detected in the screen. The authors openly state that this may be a problem, though, and I cannot offer advice on how to improve the setup.

As mentioned above, the authors need to demonstrate lysosomal mis-sorting of Atp1a3 upon knockdown/knockout/inhibition of GAK. Otherwise, all the functional evidence comes from the phosphorylation-mutants, which could potentially generate artifacts (mis-folding and aberrant ubiquitination for example).

Minor points:

Page 9: I don't think it is accurate to state that alkaline phosphatase decreases the "expression" of pT705. It decreases the phosphorylation state of said threonine residue but leaves its expression unaffected. This sentence is not the only one where the term "expression" is somewhat misplaced.

Introduction, line no. 6: "At its N-terminal"? Shouldn't this be either "At its N-terminal end" or "At its N-terminus"?

Page 13: "Residual GAK expression can likely be attributed to non-excitatory neurons such as inhibitory neurons and glia". This implies that glia are non-excitatory neurons?

Reviewer #2 (Comments to the Authors (Required)):

The manuscript by Lin et al reveals a new role of cyclin-G-associated kinase (GAK) in regulating Na/K-ATPase. It has well established the role of GAK in clathrin uncoating, but thanks to its different functional domains GAK might play multiple roles, likely different in dependence of cell context as suggested by studies based on knockout and transgenic mice. Here, the authors,

through a chemical genetic approach, identified four potential new GAK targets focusing mainly their attention to the α -subunit of Na,K-ATPase, Atp1a3.

While authors have well validated, in experiment performed in HEK 293T cells, SipaL1 and Atp1a3 as novel GAK substrates and also demonstrated the phosphorylation site in both proteins, I have some concerns on the results regarding how GAK-mediated phosphorylation might regulate the Atp1a3 trafficking and, in consequence, its activity (see below).

Overall, some major points should be addressed before considering the manuscript for publication:

1) From pictures in Figure EV1D and EV1E, it is difficult to interpret the subcellular localization of GAK. It is clear that GAK is localized in the cytoplasm preferentially in discrete structures that can resemble ER signal and are in contact to Golgi membranes. The signals are too saturated, probably due to protein overexpression. Authors should improve the quality of images (e.g., by labeling at least ER and Golgi with antibodies for endogenous proteins of these compartments) and perform a more accurate analysis of co-localization.

2) Phosphorylation assays indicate that GAK kinase domain is sufficient to phosphorylate Atp1a3, but from the gel (Fig. 3B) I have the impression that phosphorylation is lower, suggesting a less efficiency. Authors should quantify the levels of T705 phosphorylation in the different conditions. Moreover, is the GAK kinase domain able to co-immunoprecipitate with Atp1a3? This information is very important to understand how GAK regulate the activity of the pump.

3) Previous studies, performed mainly in kidney cells, showed that the surface localization of Na,K-ATPase is dependent of hierarchy of mechanisms including the interaction with actin cytoskeleton, the continuous turn over regulated by phosphorylation dependent-endocytosis. Hence, a key point is to understand whether GAK mediated phosphorylation regulates the trafficking of Atp1a3 toward the plasma membrane or from the plasma membrane or both. The data presented here support the second hypothesis, however there is no direct proof and some data are not very convincing:

a) It is important to show the localization of wt ATP1a3 and mutants (both TA and TD) in the entire neurons;

b) Authors should assay directly (by biochemical or fluorescence assays) whether the transport of Atp1a3 is impaired when GAK phosphorylation is abrogated (e.g. by using specific GAK inhibitors or by GAK silencing);

c) Pictures in Fig. EV3A do not supports what authors claim. It seems that there is a different distribution when wt ATP1a3 is tagged with HA or with GFP: HA-ATP1a3 appears more intracellularly than GFP tagged form (probably this difference is a linked to the degree of overexpression, this creates confusion).

Moreover, while it is clear that the TD mutant is retained in the cell body (faint signal in dendrites), the quality of Atp1a3 signal as that of ER staining does support their conclusion and its distribution is different in the two panels, left and right panels of Fig EV3B. What is true?

d) Fig. EV4: it seems that, with respect to wt protein, less number of TA positive puncta co-localize with TfR and there is some degree of co-localization with Rab7, supporting the hypothesis that TA mutant is targeted to lysosomal degradation instead to be recycled. It is important to perform quantitative analysis (at least the percentage of cells in which wt and TA Atp1a3 co-localize with different endosomal markers or counting the number of co-localizing puncta per cell).

4) Finally, authors identified and also validated Sipal1 as novel GAK target, and then they did not mention/comment in the discussion. These data appear uncorrelated with the rest of the manuscript. Even so, it will be important to understand which domain of GAK mediate the interaction with Sipal1 considering that only the Act1 domain of Sipal1 is sufficient to co-immunoprecipitate with GAK. The authors could test whether GAK1-400 is still able to co-immunoprecipitate with Sipal1 or should at least discuss the potential binding domain of GAK.

Minor comments

Figure legends are often not exhaustive and, in some cases, there are some mistakes:

Figure legend 1C and 1 E lacks of some information (e.g., what stand for THP? What represents the gel blotted for HA? Are GAK chimeric proteins His or HA tagged?);

Figure legend 2A: there is no proper correspondence between colors (grey, blue and red labeling the different type of proteins) in the cartoon in the Figure 2A and the text;

Figure legend EV2: What stand for 1, 2, 3? Are different samples?

Figure legend EV2B and 3A: describe that HA or myc-tagged pRK5 has been used as control of immunoprecipitation.

Reviewer #3 (Comments to the Authors (Required)):

Overall the manuscript provides a new and an important information, identifying Na⁺K⁺-ATPase alpha-subunits as GAK substrate. This phosphorylation site seems critical for Na⁺K⁺-ATPase trafficking/functioning. Authors also show that the phosphorylation is activity dependent. Considerable effort is put on this work and new tools have been generated to support their claim. Tough there is no doubt that Na⁺K⁺-ATPase is phosphorylated by GAK, the experimental evidences suggesting that the pT705 affects trafficking is not convincing (see point 8). Since the entire manuscript is built on trafficking, this part should be strengthened.

1. The last paragraph of the introduction section should be modified to clearly state their message.
2. Page 4. The text related to C190 and V99 mutations is irrelevant if these data are not being shown. Remove the text or provide the relevant data.
3. Try to write result section related to Fig 1D/E in a simplified manner. In the current form, it is complex to navigate through the text. For example, the sentence seems to be in the middle of nowhere "A key component L89, F101, H171 and F192 (Fig 1D)".
4. One problem I see is that the final GAK-mutant highly resembles AAK1 kinase. This is one of the major limitation of the screen. Since AAK1 also phosphorylates Na⁺K⁺-ATPase at the same residue, the specificity remains a question. Comment on this in discussion.
5. Show full size blot and molecular weight marker at least once for each antibody used, specifically for ATP1A3, GAK, pT705. EV1C-E is irrelevant as the authors are showing the expression of GAK in

neurons following over expression. Of course, when you transfect any protein, it will be seen in Golgi and ER. These data will be relevant if shown for endogenous GAKs.

6. Was only 4-proteins identified as GAK substrate? Authors have to provide the complete table and also deposit the proteomics data in a dataset such as Pride as per standard protocol and journal policy.

7. Fig EV2D and Fig 4A, it is said that the targets of GAK (SIPA1L1 and ATP1A3) are phosphorylated more at embryonic stages. While it is evident from the blots, provide quantifications for the same and methodological details on how much samples were loaded in each case. It appears that GAK expression is higher in adults, is this the case?

8. The experimental evidence that phospho-mutants have altered trafficking is not satisfactory. What is shown in Figure 5A is assembly in HEK cells. Why is the glycosylated form not detected in input? Cell surface biotinylation should be performed to show that the fully assembled form is indeed trafficked to the cell surface.

The qualitative-only imaging data following overexpression (Fig 5, EV3, EV4) cannot be used to make any major claims. For example, the GFP-Atp1a3 (WT) image in Figure 5B is highly saturated while the GFP-Atp1a3 (TA) is acquired/shown at very low intensity. One can easily identify Atp1a3-TA in several spines, this means they are trafficked to distant locations. Now if one looks at the GFP-Atp1a3 (WT) in Fig 5C or EV3, a punctate distribution of Atp1a3 (WT) can be seen. At this moment, it cannot be concluded that the transfected proteins are on the plasma membrane or in the cytosol. To see if they are on the membrane, the authors should use a pHluorin-tagged version. In some cases the neuron morphology is so poor that the cells are most likely in poor health. These are critical issues as currently the images are prepared as per the intended message authors want to see/show. Either these data should be completely removed or improved with proper quantifications. Authors should carefully select transfected neurons with low-expression level of exogenous proteins.

9. I did not find the cell-survival data in COS cells?

10. "We could not assess the levels of Atp1a3 pT705 as the antibody does not detect endogenous pT705 in mouse tissue". At the same time, the authors are able to detect the pT705 in rat tissue (Figure 4A). Comment on what did the authors see?

tober 29, 2018

The Francis Crick Institute Laboratory 1 Midland Road London NW1 1AT
+44 (0)203 796 0000 info@crick.ac.uk www.crick.ac.uk

29/10/2018

To the editor and reviewers:

We thank the reviewers very much for their highly useful comments on our manuscript. We have done additional experiments and analysis and strengthened our conclusions on the role of T705 phosphorylation on ATP1a3 for Na⁺/K⁺-ATPase trafficking. First, we performed surface biotinylation assays in HEK293T cells. These experiments show that the surface levels of ATP1a3 T705A mutant is highly reduced when compared to WT ATP1a3. We added a second set of experiments, where we inhibited GAK function or NAK family kinase function, using specific kinase inhibitors in COS7 cells. We then measured the surface distribution of ATP1a3 using light microscopy and automated quantification of Na⁺/K⁺-ATPase localization to the cell edge via an FIJI plugin, ADAPT. We found that when both GAK and as well as NAK family kinases AAK1 and BIKE are inhibited using a combination of two specific kinase inhibitors, there was a significant reduction in the distribution of ATP1a3 at the cell edge. These experiments indicate that localization of wild type ATP1a3 is regulated by NAK family kinases. Third, we now present new data on ATP1a3 dendritic accumulations in neuronal cultures. We show that dendritic accumulations of T705A phosphomutant ATP1a3 co-localizes with endogenous endosomal marker EEA1. We conclude that T705 phosphorylation by GAK and other NAK family members regulate the trafficking of Na⁺/K⁺-ATPase to the cell surface, preventing its accumulation in endosomes. Finally, we added a Discussion section relating our findings to previously described functions of yeast homologs of NAK family kinases, prk1p and ark1p, as well as AAK1, a member of mammalian NAK family kinases. In the context of existing literature, we discuss our findings on the redundancy of NAK family kinases and their roles in protein/membrane recycling. We think our manuscript is significantly improved by these new additions and many other clarifications and additions outlined below. We hope the referees will find that their concerns have been addressed in this version of our manuscript and we hope that they will find it suitable for publication.

Please find below our point-by-point response to reviewers' comments.

Sila Ultanir

Group Leader
Kinases and Brain Development Lab
The Francis Crick Institute,
1 Midland Road,

NW1 1AT, London, UK
Phone: + (44) 02037961613

Reviewer #1 (Comments to the Authors (Required)):

In the present study, Lin et al. employ extensive mutagenesis of the GAK kinase to enable a chemical screening approach to identify direct GAK substrates. While their screen fails to identify some known substrates, it proved useful in identifying threonine 705 of the Potassium-Sodium exchanger (NKA) subunit ATP1a3 as a potential substrate. Through the use of phosphorylation resistant and mimetic mutants, the authors demonstrate that the phosphorylation of T705 is required for plasma membrane delivery and overall functionality of NKA. While most of the data is convincing, some important evidence is missing in the manuscript: The authors need to show that loss of GAK in a cellular system perturbs ATP1a3 trafficking and thus NKA function. They provide some indirect evidence from conditional knockout mice but I couldn't find direct evidence that loss of GAK results in a loss of ATP1a3 phosphorylation, trafficking and ultimately NKA function. I suggest to use their COS-7-Ouabain lethality model to test whether GAK knockdown (or knockout with Crispr/Cas9, or inhibitor treatment) also compromises NKA trafficking and function. As it is, most of the functional evidence stems from the phosphorylation mutant proteins, which leaves this reviewer slightly skeptical. Besides this criticism, I am overall supportive of publication as the manuscript is very extensive, provides a significant advance and mostly convincing evidence. Some additional concerns are listed under major and minor points.

Major points:

1.) The authors show that the phosphorylation resistant TA mutant of Atp1a3 is being mis-sorted into lysosomes. It would be good to confirm this with an antibody against endogenous LAMP1 (or LAMP2 or CD63). At least in our hands, Lamp1-GFP is not a good marker when overexpressed as it is also subject to trafficking to and from lysosomes.

We thank the reviewer for this important comment. In our revised manuscript, we present endogenous immunostainings for the early endosome marker EEA1 and an lysosomal marker lamp1, using well-established antibodies. Indeed, we observed a small amount of colocalization of intracellular WT or phosphomutant (T705A) ATP1a3 accumulations with lamp1, after lysosomal blockage as described in our manuscript (now shown in Fig S5C). There was also no difference between percent colocalizations between WT and ATP1a3 T705A, (the quantifications of lamp1 co-localizations are stated in Figure legends of Fig S5C (9% and 8% puncta colocalized with lamp1 for WT and ATP1a3 TA, respectively). Interestingly, we found substantial co-localizations of intracellular ATP1a3 puncta with endogenous EEA1, 25 ± 14 % of total puncta for wt and 39 ± 8 % of total puncta for TA. Our new data on EEA1 co-localization is reported in Fig 6B and in the Results section. In the light of these new findings and further data described below, we altered our

conclusions to state that phosphorylation of ATP1a3 at T705 is necessary for its efficient exit from early endosomes to recycle to the plasma membrane. We again conclude that phosphorylation of the alpha subunit of NKA is necessary for its localization to plasma membrane and overall functionality.

2.) The authors use a heavily truncated fragment of GAK-AS for their substrate screen. Could this distort the substrate specificity? It is somewhat concerning that the known substrates AP-2 and PP2A, which should be present in brain lysates, was not detected in the screen. The authors openly state that this may be a problem, though, and I cannot offer advice on how to improve the setup.

We have stated this openly now in the results section on “Chemical genetic identification of GAK substrates” saying “Neither AP2 μ 2 nor PP2A B γ appeared in our list of candidate substrates, which could be due to either low abundance, or that using GAK¹⁻⁴⁰⁰ instead of full length GAK resulted in less effective interaction with binding partners. In addition, due to their biophysical properties some peptides may not be detected by mass spectrometry. Therefore, not detecting a phosphorylation site would not necessarily mean that it is not phosphorylated.”

3.) As mentioned above, the authors need to demonstrate lysosomal mis-sorting of Atp1a3 upon knockdown/knockout/inhibition of GAK. Otherwise, all the functional evidence comes from the phosphorylation-mutants, which could potentially generate artifacts (mis-folding and aberrant ubiquitination for example).

We thank the reviewer for these comments. In our first version, we showed before that ATP1a3 can be phosphorylated by AAK1 in addition to GAK (Fig 3D). Mammalian NAK kinase family also includes MPSK1 and BIKE, kinases as mentioned in “Introduction”. AAK1 and GAK can both phosphorylate the μ 2 subunit of AP-2. It is possible that similarly, other substrates of NAK family kinases would be shared between members of this kinase family. In yeast, two homologs of NAK family kinases would need to be simultaneously deleted to observe major effects {Sekiya-Kawasaki, 2003 #676} (now included in our the new version of our manuscript) on actin cytoskeleton at endocytic zones, indicating that redundancy of NAK family kinase functions could be evolutionarily very conserved. The yeast homologs, their function and redundancy is now discussed in discussion section:

“NAK family kinases are conserved from yeast to mammals. In yeast, loss-of-function mutations of the NAK family kinases *prk1p* and *ark1p* cause accumulations of actin-associated endocytic vesicles that cannot traffic further to other membrane compartments {Sekiya-Kawasaki, 2003 #676}. In mammals, AAK1 regulates endocytic trafficking {Conner, 2002 #527} and trafficking of cell surface receptors. AAK1 regulates notch receptor recycling and signalling {Gupta-Rossi, 2011 #672}, and neuregulin-1/ ErbB4 signaling by altering ErbB4 trafficking {Kuai, 2011 #671}. In our loss of function analysis enabled by specific GAK and AAK1/BIKE inhibitors, we find that there is a significant reduction in NKA’s peripheral distribution in COS7 cells. This finding is in agreement with reduced phosphomutant surface expression and increased presence of NKA in intracellular compartments. Therefore, it is possible that multiple members of NAK family including GAK and AAK1 could play complementary roles in receptor recycling via phosphorylating cargo proteins.”

In order to test if inhibition of NAK family kinases also results in trafficking defects of the Na/K pump / NKA, we used previously described inhibitors of GAK (12i) and AAK1 and BIKE (LP) in COS7 cells. We measured subcellular distribution of HA tagged ATP1a3 by using an ImageJ plugin (ADAPT), which can render the cell outline and calculate the intensities as a function of distance from cell center to periphery. We find that dual application of inhibitors caused a reduction in the percentage of total WT ATP1a3 that is distributed to the cell's edge, indicating that surface localization of NKA is affected by inhibition of NAK family kinases. These new experiments and analysis are now shown in Figure 5B-D.

Minor points:

1.) Page 9: I don't think it is accurate to state that alkaline phosphatase decreases the "expression" of pT705. It decreases the phosphorylation state of said threonine residue but leaves its expression unaffected. This sentence is not the only one where the term "expression" is somewhat misplaced.

We have gone through the text and altered "expression" to "amount" or "levels" where appropriate or deleted the word altogether.

2.) Introduction, line no. 6: "At its N-terminal"? Shouldn't this be either "At its N-terminal end" or "At its N-terminus"?

Very true; this has been changed to "At its N-terminus".

3.) Page 13: "Residual GAK expression can likely be attributed to non-excitatory neurons such as inhibitory neurons and glia". This implies that glia are non-excitatory neurons?

Thank you for catching that; the text has now been changed to read, "Residual GAK expression can likely be attributed to glia, or to non-excitatory neurons such as inhibitory neurons."

Reviewer #2 (Comments to the Authors (Required)):

The manuscript by Lin et al reveals a new role of cyclin-G-associated kinase (GAK) in regulating Na/K-ATPase. It has well established the role of GAK in clathrin uncoating, but thanks to its different functional domains GAK might play multiple roles, likely different in dependence of cell context as suggested by studies based on knockout and transgenic mice. Here, the authors, through a chemical genetic approach, identified four potential new GAK targets focusing mainly their attention to the α -subunit of Na,K-ATPase, Atp1a3. While authors have well validated, in experiment performed in HEK 293T cells, Sipal1 and Atp1a3 as novel GAK substrates and also demonstrated the phosphorylation site in both proteins, I have some concerns on the results regarding how GAK-mediated phosphorylation might regulate the Atp1a3 trafficking and, in consequence, its activity (see below).

Overall, some major points should be addressed before considering the manuscript for publication:

1) From pictures in Figure EV1D and EV1E, it is difficult to interpret the subcellular localization of GAK. It is clear that GAK is localized in the cytoplasm preferentially in discrete structures that can resemble ER signal

and are in contact to Golgi membranes. The signals are too saturated, probably due to protein overexpression. Authors should improve the quality of images (e.g., by labeling at least ER and Golgi with antibodies for endogenous proteins of these compartments) and perform a more accurate analysis of co-localization.

Overexpressed GAK localized throughout the cytoplasm as well as Golgi. We lack a good GAK antibody for endogenous GAK immunostaining. We decided that description of overexpressed GAK is not essential for our manuscript and as per the comment #5 by Reviewer 3, thus we have taken out EV1 panels C-E.

2) Phosphorylation assays indicate that GAK kinase domain is sufficient to phosphorylate Atp1a3, but from the gel (Fig. 3B) I have the impression that phosphorylation is lower, suggesting a less efficiency. Authors should quantify the levels of T705 phosphorylation in the different conditions.

Moreover, is the GAK kinase domain able to co-immunoprecipitate with Atp1a3? This information is very important to understand how GAK regulate the activity of the pump.

The different conditions in Figure 3B have now been quantified and charts have been added to the Figure 3B. No differences were observed in pT705 levels between GAK full length and GAK kinase domain.

The GAK kinase domain is able to co-IP Atp1a3. A figure illustrating this has been added to Fig 3A (right panel) and the text and figure legend has been modified accordingly.

3) Previous studies, performed mainly in kidney cells, showed that the surface localization of Na,K-ATPase is dependent of hierarchy of mechanisms including the interaction with actin cytoskeleton, the continuous turn over regulated by phosphorylation dependent-endocytosis. Hence, a key point is to understand whether GAK mediated phosphorylation regulates the trafficking of Atp1a3 toward the plasma membrane or from the plasma membrane or both. The data presented here support the second hypothesis, however there is no direct proof and some data are not very convincing:

We thank the reviewer for this thoughtful comment. Indeed, it is our hope to decipher how phosphorylation of ATP1a3 alters its trafficking in a detailed manner. We have now completed additional set of experiments as outlined in our rebuttal and we hope that this would provide significant progress in our understanding of phospho-regulation of cargo trafficking.

a) It is important to show the localization of wt ATP1a3 and mutants (both TA and TD) in the entire neurons;

We agree with the reviewer's comment. We have now have included full images of neurons expressing WT, TA and TD mutant ATP1a3 in the new Fig S6. The same neurons are shown at higher magnification in Fig 6B. We recommend that the reviewers examine these images carefully by zooming in at high magnification. We believe that addition of Fig S6 (which contains 2 separate lower magnification images of neurons) and addition of Fig 6B, which contains more detailed examination of high magnification dendrites with colocalization of GFP cell fill and EEA1, will provide the needed additional support for the striking differences in distribution of ATP1a3 mutants. We did not take away any of the previous dataset images, which are presented in Fig 6A (GFP tagged ATP1a3) and Fig S5A (HA-tagged ATP1a3).

b) Authors should assay directly (by biochemical or fluorescence assays) whether the transport of Atp1a3 is

impaired when GAK phosphorylation is abrogated (e.g. by using specific GAK inhibitors or by GAK silencing);

We thank the reviewer for this important comment. Using inhibitors targeting GAK and NAK family kinases AAK1/BIKE, we now report that inhibition of these kinases lead to altered trafficking of ATP1a3 in COS7 cells.

Below is the same as response to Reviewer 1, comment 3:

We note that ATP1a3 can be phosphorylated by AAK1 in addition to GAK. Mammalian NAK kinase family also includes MPSK1 and BIKE, kinases as mentioned in "Introduction". Both AAK1 and GAK can phosphorylate the μ 2 subunit of AP-2. Therefore, it is possible that, other substrates of NAK family kinases would be shared between members of this kinase family. In yeast, two homologs of NAK family kinases would need to be simultaneously deleted to observe major effects (ref) on actin cytoskeleton at endocytic zones, indicating that redundancy of NAK family kinase functions could be evolutionarily very conserved.

In order to test if inhibition of NAK family kinases also results in trafficking defects of NKA, we used previously described inhibitors of GAK (12i) and AAK1 and BIKE (LP) in COS7 cells. We measured subcellular distribution of HA tagged ATP1a3 by using an ImageJ plugin (ADAPT) which can render the cell outline and calculate the intensities as a function of distance from cell center to periphery. We find that dual application of inhibitors caused a reduction in the percentage of total WT ATP1a3 that is distributed to the cell's edge, indicating that surface localization of NKA is reduced by inhibition of NAK family kinases. Due to redundancy within NAK kinase family we think additional kinase inhibitors (also targeting MPSK1) might be required to obtain further alterations in NKA trafficking.

c) Pictures in Fig. EV3A do not supports what authors claim. It seems that there is a different distribution when wt ATP1a3 is tagged with HA or with GFP: HA-ATP1a3 appears more intracellularly than GFP tagged form (probably this difference is a linked to the degree of overexpression, this creates confusion). Moreover, while it is clear that the TD mutant is retained in the cell body (faint signal in dendrites), the quality of Atp1a3 signal as that of ER staining does support their conclusion and its distribution is different in the two panels, left and right panels of Fig EV3B. What is true?

The distribution of Atp1a3 WT when tagged with HA vs GFP has a slight difference, this is mainly because the HA tag is amplified using immunostaining against HA tag. This process generates more background for HA staining when compared to GFP tagged ATP1a3, which is not further amplified for immunostaining. These images are now shown in Fig S5A (HA tagged ATP1a3) and Fig 6A (GFP tagged ATP1a3). We further added more images of HA tagged ATP1a3 (including low magnifications) in Fig S6. We believe that including both GFP tagged and HA tagged ATP1a3 corroborates our findings. With new data as well as analysis reported in Results section "Phosphomutant ATP1a3 Accumulates in Endosomal Compartments in Neurons" we think the characterization of the localization of ATP1a3 and its mutant forms in neurons is now much better described. The higher magnification images of neurons in Fig S6 are shown in Fig 6B.

The left and right panels of the first version's Fig EV3B (now Fig S5B) are optical slices as mentioned in the figure legend ("3-5 stacked confocal images"). This was done to best show the overlay of Atp1a3 TD with either the ER or the Golgi without masking the expression. Below is the max projection of Atp1a3 TD for both cells, which appear similar.

d) Fig. EV4: it seems that, with respect to wt protein, less number of TA positive puncta co-localize with TfR and there is some degree of co-localization with Rab7, supporting the hypothesis that TA mutant is targeted to lysosomal degradation instead to be recycled. It is important to perform quantitative analysis (at least the percentage of cells in which wt and TA Atp1a3 co-localize with different endosomal markers or counting the number of co-localizing puncta per cell).

We counted the number of Atp1a3 WT or TA puncta that co-localized with the different endosomal markers per a 50 μm dendritic segment (from 3-5 individual neurons). The average percentage of co-localization has been added to the Results section “Phosphomutant ATP1a3 Accumulates in Endosomal Compartments in Neurons”. The percentage of co-localizing puncta with each endosomal marker was not changed between WT and TA.

We show that there is a substantial (4 folds) increase in dendritic puncta in ATP1a3 TA expressing neurons when compared to ATP1a3 WT expressing neurons. These indicate an accumulation of endosomal ATP1a3 in T705A mutants. We further show that a 39% of these puncta co-localize with endogenous EEA1, an early/sorting endosome marker. We propose that the trafficking of ATP1a3 from sorting endosomes to the cell surface is impaired in ATP1a3 TA mutants.

4) Finally, authors identified and also validated SipaL1 as novel GAK target, and then they did not mention/comment in the discussion. These data appear uncorrelated with the rest of the manuscript. Even so, it will be important to understand which domain of GAK mediate the interaction with SipaL1 considering that only the Act1 domain of SipaL1 is sufficient to co-immunoprecipitate with GAK. The authors could test whether GAK1-400 is still able to co-immunoprecipitate with SipaL1 or should at least discuss the potential binding domain of GAK.

We agree with the reviewer’s comments. Indeed, we believe that cargo phosphorylation by NAK family kinases is an important component of membrane/ receptor recycling process. We decided to include Sipa111 data in our manuscript to point out that Sipa111 as a substrate in this context, which could initiate future studies. In this context we have included the following text in the Discussion section “In conjunction with known functions of NAK family kinases in membrane trafficking, we propose a simple model whereby GAK phosphorylates cargo proteins such as ATP1a3 and Sipa111 during or following endocytosis and this phosphorylation is essential for efficient progression of ATP1a3 to subsequent trafficking steps, particularly for recycling back to plasma membrane (Fig 8A). Our study demonstrates that without NAK family kinases or Atp1a3 T705 phosphorylation, NKA trafficking and function are impaired.”

Minor comments

Figure legends are often not exhaustive and, in some cases, there are some mistakes:

1.) Figure legend 1C and 1E lacks of some information (e.g., what stand for THP? What represents the gel blotted for HA? Are GAK chimeric proteins His or HA tagged?);

Figure legend 1C has been re-written to include the information that the chimeric proteins are HA-tagged (this has also been indicated in the figure image itself for improved clarity) and a definition of THP has been provided in the legend as well. However, we have not repeated the information from 1C in 1E, as it may be seen as redundant.

2.) Figure legend 2A: there is no proper correspondence between colors (grey, blue and red labeling the different type of proteins) in the cartoon in the Figure 2A and the text;

References to what the colours in the Figure and Legend represent have been added to the text for clarification – thank you for catching that!

3.) Figure legend EV2: What stand for 1, 2, 3? Are different samples?

Yes, these are different samples. The following text “Numbers 1-3 represent independent samples from different brains” has been added to the figure legend to clarify this.

4.) Figure legend EV2B and 3A: describe that HA or myc-tagged pRK5 has been used as control of immunoprecipitation.

The following text has been added to both legends: “HA-tagged and myc-tagged pRK5 vectors have been used as IP controls”.

Reviewer #3 (Comments to the Authors (Required)):

Overall the manuscript provides a new and an important information, identifying Na⁺K⁺-ATPase alpha-subunits as GAK substrate. This phosphorylation site seems critical for Na⁺K⁺-ATPase trafficking/functioning. Authors also show that the phosphorylation is activity dependent. Considerable effort is put on this work and new tools have been generated to support their claim. Though there is no doubt that Na⁺K⁺-ATPase is phosphorylated by GAK, the experimental evidences suggesting that the pT705 affects trafficking is not convincing (see point 8). Since the entire manuscript is built on trafficking, this part should be strengthened.

1. The last paragraph of the introduction section should be modified to clearly state their message.

We thank the reviewer for this comment. We now re-wrote the last paragraph of the introduction section more carefully, clarifying our findings.

2. Page 4. The text related to C190 and V99 mutations is irrelevant if these data are not being shown.

Remove the text or provide the relevant data.

The relevant data has been provided as Figure S1.

3. Try to write result section related to Fig 1D/E in a simplified manner. In the current form, it is complex to navigate through the text. For example, the sentence seems to be in the middle of nowhere "A key component L89, F101, H171 and F192 (Fig 1D)".

We have rearranged and tried to simplify the text as follows and hope that it is more clear: "A key component known to affect kinase activity is the regulatory spine, which consists of four hydrophobic residues spanning the tertiary structure of the kinase domain (Kornev et al., 2006). The introduction of Isoleucine, a hydrophobic β -branched amino acid, at a -2 position with respect to the gatekeeper residue located on the β 5 strand (Fig 1D) is able to compensate for the loss of hydrophobicity caused by gatekeeper mutation (Joseph and Andreotti, 2011). In GAK, the hydrophobic spine residues are L89, F101, H171 and F192 (Fig 1D)."

4. One problem I see is that the final GAK-mutant highly resembles AAK1 kinase. This is one of the major limitation of the screen. Since AAK1 also phosphorylates Na⁺K⁺-ATPase at the same residue, the specificity remains a question. Comment on this in discussion.

We thank the reviewer for this important comment. We note that in overexpression experiments in HEK293 cells, where wild type GAK or wild type AAK1 is co-expressed with ATP1a3 and probed by the ATP1a3 phosphoT705 antibody indicates that both wild type kinases are able to phosphorylate these substrates (in fact GAK phosphorylates more efficiently). However, NAK family kinases are known to have overlapping substrates, as in the case of the μ 2 subunit of AP-2. We added the following text to Discussion "Importantly, due to overlapping nature of substrate specificity among NAK family kinases, we cannot conclude that the identified putative substrates (Table 1) are phosphorylated by GAK and/ or other NAK family kinases in organisms. In addition, all putative phosphorylation sites need to be validated by kinase assays."

5. Show full size blot and molecular weight marker at least once for each antibody used, specifically for ATP1A3, GAK, pT705. EV1C-E is irrelevant as the authors are showing the expression of GAK in neurons following over expression. Of course, when you transfect any protein, it will be seen in Golgi and ER. These data will be relevant if shown for endogenous GAKs.

We thank the reviewer for these comments. EV1C-E have been removed and full size blots are provided as source data.

6. Was only 4-proteins identified as GAK substrate? Authors have to provide the complete table and also deposit the proteomics data in a dataset such as Pride as per standard protocol and journal policy.

A complete table of Threonine phosphorylated GAK substrates is provided as Table 1. In the older version of Figure 1 we had highlighted the candidates that were linked to human disorders. We have now removed this information from Fig 1 and for clarity we are only providing the complete table (Table 1). The criterion of inclusion in this table is now stated in the results section:

"From the data generated, only phospho-peptides detected at least three times in GAK AS samples and not in any GAK KD negative controls are included in a list of substrate candidates. Of the potential GAK substrates identified, there appears to be a predominant preference for a small TG motif, where T is phosphorylated Threonine and G is Glycine (Fig 2A). This motif matches with that of a previously identified GAK and AAK1

substrate, the T156 residue of the AP-2 adaptor complex μ 2-subunit (SQITSQVTGQIGWRR) (Korolchuk & Banting, 2002, Ricotta, Conner et al., 2002, Zhang et al., 2005b), as well as the GAK substrate protein phosphatase 2A (PP2A) B γ T104 residue (SNPTGAEFDP) (Naito et al., 2012). Therefore, in the list of putative GAK substrates we removed the three candidates that do not contain the "TG" motif (Table 1)." The proteomics data has been deposited in Pride, with dataset identifier number PXD011319.

7. Fig EV2D and Fig 4A, it is said that the targets of GAK (SIPA1L1 and ATP1A3) are phosphorylated more at embryonic stages. While it is evident from the blots, provide quantifications for the same and methodological details on how much samples were loaded in each case. It appears that GAK expression is higher in adults, is this the case?

Quantifications have been provided within the text and in figure legends, and methodology has been added to Materials and Methods under the heading Western blotting. GAK appears slightly elevated in adult animal brains but this is not significant when compared to embryonic levels.

8. The experimental evidence that phospho-mutants have altered trafficking is not satisfactory. What is shown in Figure 5A is assembly in HEK cells. Why is the glycosylated form not detected in input? Cell surface biotinylation should be performed to show that the fully assembled form is indeed trafficked to the cell surface.

The glycosylated version is detectable in input, but only after increased exposure times at which point other proteins have been overexposed. We have performed surface biotinylation experiments and included the information as Fig S4. Surface biotinylation has been added to Methods section. We show that surface ATP1a3 is substantially reduced in ATP1a3 TA and ATP1a3 TD mutants with both beta1 and beta2 subunit co-expression in HEK293 cells. Our results were quantified and these are presented in FigS4. These new data indicate that indeed the surface expression of ATP1a3 is affected by phosphorylation at T705.

The qualitative-only imaging data following overexpression (Fig 5, EV3, EV4) cannot be used to make any major claims. For example, the GFP-Atp1a3 (WT) image in Figure 5B is highly saturated while the GFP-Atp1a3 (TA) is acquired/shown at very low intensity. One can easily identify Atp1a3-TA in several spines, this means they are trafficked to distant locations. Now if one looks at the GFP-Atp1a3 (WT) in Fig 5C or EV3, a punctate distribution of Atp1a3 (WT) can be seen. At this moment, it cannot be concluded that the transfected proteins are on the plasma membrane or in the cytosol. To see if they are on the membrane, the authors should use a pHluorin-tagged version. In some cases the neuron morphology is so poor that the cells are most likely in poor health. These are critical issues as currently the images are prepared as per the intended message authors want to see/show. Either these data should be completely removed or improved with proper quantifications. Authors should carefully select transfected neurons with low-expression level of exogenous proteins.

We thank the reviewer for these comments. First of all, we believe that our neuronal morphologies are very healthy. The pictures we used in the first version of our manuscript may not have reflected our observations faithfully. In order to clarify the overall morphologies of our neurons we now included zoomed out images that contain all of the neuronal dendritic arbor of neurons expressing HA-tagged ATP1a3 (WT, TA and TD versions) in Fig S6A. In Fig S6B we include slightly more zoomed in versions that show the dendrites and spines as well as the ATP1a3 HA. An even higher magnification of the same neurons (for WT and TA) are shown in Fig 6B, where HA staining and GFP cell fill are colocalized with endogenous EEA1 staining. We believe that these set of images now clarify that the overall morphologies of our neurons were very healthy.

The second issue raised by the reviewer was the way the images are represented. It is important to note that images in first version Fig 5B (now Fig 6A) were taken at the same laser power and gain. These images describe the localization of GFP-Atp1a3 WT and TA. These images reflect a major difference in trafficking – ATP1a3 TA it is not in spines, but is present in accumulations in dendrites and the cell body.

As stated in response to Reviewer 2, Comment 3C:

“The distribution of Atp1a3 WT when tagged with HA vs GFP has a slight difference, this is mainly because the HA tag is amplified using immunostaining for the HA tag. This process generates more background for HA staining when compared to GFP tagged ATP1a3, which is not further amplified for immunostaining. These images are now shown in Fig S5A (HA tagged ATP1a3) and Fig 6A (GFP tagged ATP1a3). We further added more images of HA tagged ATP1a3 (including low magnifications) in Fig S6. We believe that including both GFP tagged and HA tagged ATP1a3 corroborates our findings. With new data as well as analysis reported in Results section “Phosphomutant ATP1a3 Accumulates in Endosomal Compartments in Neurons” we think the characterization of the localization of ATP1a3 and its mutant forms in neurons is now much better described. The higher magnification images of neurons in Fig S6 are shown in Fig 6B.”

We agree with the reviewer that only qualitative descriptions would not be sufficient for reporting our findings. We have now measured the number of intracellular ATP1a3 WT or TA accumulations in dendrites. We find that there is about 4 folds increase in the number of puncta per dendrite length (imaging dendrites with similar diameters). In addition, we also quantified the colocalization percentages between ATP1a3 puncta and the endogenous lamp1 and EEA1 as well as fluorescent tagged markers of endosomal compartments. We show a substantial increase in the intracellular puncta formation in ATP1a3 TA mutant.

These quantifications are reported in Results section “Phosphomutant ATP1a3 Accumulates in Endosomal Compartments in Neurons”. The results section that describes the quantifications is copied below “To determine at which step of the endocytic trafficking pathway Atp1a3 TA-containing pumps were being retained, we used endogenous stainings or co-transfections with different intracellular membrane compartment markers in rat hippocampal neurons expressing Atp1a3 WT and TA. We found that a subset of puncta in WT and in TA mutants colocalize with endogenous EEA1 (Fig 6B, lower magnifications of these neurons are shown in Fig S6A&B). We quantified the number of ATP1a3 puncta along the length of dendrites, imaging dendrites with comparable t. The density of intracellular ATP1a3 accumulations in dendrites was significantly higher in TA (approximately 4 folds (0.45 ± 0.1 puncta/ μm) when compared to WT (0.11 ± 0.04 puncta/ μm) ($p = 0.017$ $n = 7$ cells each). However, the percentage that co-localizes with EEA1 was not significantly different (25 ± 14 % for wt and 39 ± 8 % for TA for the same group). In addition, we found that co-transfections of Atp1a3 WT or Atp1a3 TA with markers for early endosomes (mCh-Rab5), recycling endosomes (mCh-Transferrin receptor; mCh-TfR) and late endosomes (mCh-Rab7a) all showed some degree of co-localization, indicating that Atp1a3 trafficking is not fully dependent upon T705 phosphorylation (Fig S7A). When the percentage of Atp1a3 WT or TA puncta co-localization with these endosomal markers was quantified, there were no differences observed in mCh-Rab5 (in %: WT 21.7 ± 4.2 ($n = 3$ cells); TA 24.3 ± 8.3 ($n = 5$ cells); $p = 0.83$), mCh-Rab7a (in %: WT 44.1 ± 6.4 ; TA 43.9 ± 10.5 ; $p = 0.98$, $n = 5$ cells, each) or mCh-TfR (in %: WT 50.8 ± 7.9 ; TA 31.3 ± 8.2 ; $p = 0.13$, $n = 5$ cells, each). In agreement with endogenous Early endosome antigen 1 (EEA1) stainings, these data indicate that similar percentage of ATP1a3 containing intracellular compartments colocalize with endocytic pathway markers. However, significantly more intracellular puncta are found in the TA mutant, specifically. Collectively, these data suggest that if Atp1a3 T705 is unable to be phosphorylated, the NKA subsequently cannot be efficiently redirected from recycling compartments to the plasma membrane, supporting its functional importance.”

9. I did not find the cell-survival data in COS cells?

We did not have a method of quantification for this particular experiment as there was no cell survival in OuAR TA and TD mutants; cells were not able to survive to even a first passage after ouabain treatment. This experiment was based on a similar one performed in a paper by Clapcote et al., 2009, which we list in our references – they too were unable to quantify this particular phenotype (mentioned under Supporting Information) although in their case, cells survived up to 3 weeks whereas with our mutants, 1 week was the limit.

10. "We could not assess the levels of Atp1a3 pT705 as the antibody does not detect endogenous pT705 in mouse tissue". At the same time, the authors are able to detect the pT705 in rat tissue (Figure 4A). Comment on what did the authors see?

We did not detect any specific signals at the right molecular weight with mouse tissue – there were only very faint non-specific bands seen even at very long exposure times.

November 15, 2018

RE: Life Science Alliance Manuscript #LSA-2018-00118-TR

Dr. Sila Ultanir
The Francis Crick Institute
1 Midland Road
London, London NW1 1AT
United Kingdom

Dear Dr. Ultanir,

Thank you for submitting your revised manuscript entitled "Chemical Genetic Identification of GAK Substrates Reveals its Role in Regulating Na⁺/K⁺-ATPase". As you will see, the reviewers appreciate the introduced changes, and we are happy to accept your manuscript in principle for publication here, pending that you address the remaining reviewer comments by a more balanced discussion and changes to the text. Note that reviewer #1 is disappointed that you didn't use a knock-out approach to better support a role for GAK in Na⁺/K⁺-ATPase trafficking. It should therefore be acknowledged in the text that a KO approach would have added further proof to your hypothesis.

While revising your manuscript, please also pay attention to the following:

- please upload individual figure files for main and suppl. figures
- please provide the main text as a docx file
- please provide the tables either as docx or excel file (they can remain in the manuscript docx file if that's easiest)
- please note that you currently mention Fig2C in the text, but Fig2C doesn't exist
- please indicate in FigS5A the location of the spines displayed in the magnifications
- please link your ORCID iD to your profile, you should have received an email with instructions of how to do so

A. FINAL FILES:

-- High-resolution figure, supplementary figure and video files uploaded as individual files: See our detailed guidelines for preparing your production-ready images, <http://life-science->

alliance.org/authorguide

B. MANUSCRIPT ORGANIZATION AND FORMATTING:

Full guidelines are available on our Instructions for Authors page, <http://life-science-alliance.org/authorguide>

Sincerely,

Reviewer #1 (Comments to the Authors (Required)):

My main concern was that the claim of GAK regulated trafficking of ATP1A3 was based solely on the phospho-site mutant protein. The authors have added some data that supports their claim that the trafficking of ATP1A3 is regulated by GAK as the dual use of two inhibitors mimicks the effect of the phospho-site mutant ATP1A3. I am not completely convinced here as the use of two inhibitors may induce significant off-target effects. A knockdown/KO of GAK would have been much preferable but I realize that I suggested the use of either knockdown and/or inhibitor use and the authors took the latter approach. If the authors are convinced that GAK promotes the export of ATP1A3 from early endosomes they could go ahead and publish this. Some doubts remain on my side.

They also added data on trafficking in neurons, which I find difficult to assess due to my lack of experience with such a system. From the other reviewers comments I derived that there is more expertise on neuronal trafficking so maybe those reviewers can provide input on the added data from neurons.

As stated above, this manuscript provides a significant advance and ATP1A3 trafficking is likely regulated by GAK but it could have been proven in a more convincing way. I believe that a lot of readers will also be somewhat skeptical of the dual use of two inhibitors and a phospho-site mutant protein to demonstrate effects on ATP1A3 trafficking. I don't really understand why no knockdown/Knockout approach of GAK was taken anywhere in the manuscript. From my side, it is up to the authors and the editor to decide whether this can be published in its current state.

Reviewer #2 (Comments to the Authors (Required)):

Authors replied in full to all concerns, therefore the manuscript is now acceptable for publication.

Reviewer #3 (Comments to the Authors (Required)):

The manuscript by Lin, Gill et al., provides a novel and important mechanism that regulates cell-surface trafficking of alpha-subunit of NKA. The reviewer recommends it for publication with some minor comments:

- 1) It should be highlighted clearly that all the alpha-subunits are GAK substrate.
- 2) There is no doubt that TA/TD mutations impair trafficking but careful interpretation is needed. a) The low-MW, non-glycosylated form of beta-subunits are trafficked to the plasma membrane (Fig S4), this means authors cannot claim that only the glycosylated forms are on cell-surface. b) TD-mutant is getting retained in the ER, at the same time they do observe that it is being trafficked to the cell-surface along with the beta-subunits. In the same lines, if TD-NKA was ER retained, why a diffused pattern is seen throughout the neuron (Fig 7). What is the evidence that TD is indeed a phospho-mimetic and not a more potent version of phospho-mutant TA? The claim that TD is ER-retained should be revisited.
- 3) What is the evidence that NKA is recycled? Because authors observe co-localization of TA mutants with various compartment of endosomal pathway, it cannot be assumed that entire NKA trafficking pathway is dependent on this-specific phosphorylation as in the last Figure.

06/12/2018

To the editor and reviewers:

Reviewer #1 (Comments to the Authors (Required)):

My main concern was that the claim of GAK regulated trafficking of ATP1A3 was based solely on the phospho-site mutant protein. The authors have added some data that supports their claim that the trafficking of ATP1A3 is regulated by GAK as the dual use of two inhibitors mimicks the effect of the phospho-site mutant ATP1A3. I am not completely convinced here as the use of two inhibitors may induce significant off-target effects. A knockdown/KO of GAK would have been much preferable but I realize that I suggested the use of either knockdown and/or inhibitor use and the authors took the latter approach. If the authors are convinced that GAK promotes the export of ATP1A3 from early endosomes they could go ahead and publish this. Some doubts remain on my side.

They also added data on trafficking in neurons, which I find difficult to assess due to my lack of experience with such a system. From the other reviewers comments I derived that there is more expertise on neuronal trafficking so maybe those reviewers can provide input on the added data from neurons.

As stated above, this manuscript provides a significant advance and ATP1A3 trafficking is likely regulated by GAK but it could have been proven in a more convincing way. I believe that a lot of readers will also be somewhat skeptical of the dual use of two inhibitors and a phospho-site mutant protein to demonstrate effects on ATP1A3 trafficking. I don't really understand why no knockdown/Knockout approach of GAK was taken anywhere in the manuscript. From my side, it is up to the authors and the editor to decide whether this can be published in its current state.

We thank the reviewer for these comments. I am disappointed that we did not provide a clear explanation in our previous response to the reviewers' comments about why we chose the inhibitor method. Although shRNA mediated knockdowns do have their limitations due to insufficient knockdown efficiency and off-target effects, in principle, we agree that genetic deletion of kinase(s) would be a preferred manipulation for loss of kinase activity. In this case, we chose to not employ the knockdown method, because multiple kinases within the NAK family may be responsible for the phosphorylation of ATP1a3. Our experiments show that that at least GAK and AAK1 are two kinases that can phosphorylate this site in cells, however other two members of NAK family, BIKE and MPSK, may also take part. We and others previously showed that GAK and AAK1 are expressed in the brain, in addition MPSK is also shown to be present in the brain, based on Allen Brain Atlas in situ hybridizations. For following up on our HEK293 assays we would need to knock-down all relevant kinases in HEK293 cells, likely a toxic combination, as NAK family kinases are essential and conserved from yeast. Instead of knocking down all and/or a subset of these kinases, we chose to use a combination of two inhibitors, which inhibit NAK family kinases. Our acute inhibitor treatment for several hours would lead to a reduction in NAK kinase activity, allowing us to evaluate changes in ATP1a3 trafficking, potentially be less toxic than shRNA combinations. We have now mentioned that a knockdown approach would have added further support for the role of NAK family kinases in Na⁺/K⁺-ATPase trafficking in Discussion section.

“Future experiments involving knockdown of GAK, AAK1 and possibly other NAK family members in neuronal cultures followed by evaluation of ATP1a3 trafficking would provide further support for their role in ATP1a3 trafficking.”

Reviewer #2 (Comments to the Authors (Required)):

Authors replied in full to all concerns, therefore the manuscript is now acceptable for publication.

Thank you very much.

Reviewer #3 (Comments to the Authors (Required)):

The manuscript by Lin, Gill et al., provides a novel and important mechanism that regulates cell-surface trafficking of alpha-subunit of NKA. The reviewer recommends it for publication with some minor comments:

1) It should be highlighted clearly that all the alpha-subunits are GAK substrate.

Thank you very much for reviewer's comments.

Following change was made in Results section “We chose to study further two of these substrates due to their important functions in neurons: Sipa1L1, which regulates dendritic spine morphology and the catalytic α -subunits of the sodium potassium pump (Atp1a1/ATP1a2/ATP1a3), which maintains resting membrane potential. The phosphorylation site we identified was conserved among α -subunit subtypes, as ATP1a3 is the most abundant alpha subunit in neurons, we focused our attention to ATP1a3.”

2) There is no doubt that TA/TD mutations impair trafficking but careful interpretation is needed. a) The low-MW, non-glycosylated form of beta-subunits are trafficked to the plasma membrane (Fig S4), this means authors cannot claim that only the glycosylated forms are on cell-surface. b) TD-mutant is getting retained in the ER, at the same time they do observe that it is being trafficked to the cell-surface along with the beta-subunits. In the same lines, if TD-NKA was ER retained, why a diffused pattern is seen throughout the neuron (Fig 7). What is the evidence that TD is indeed a phospho-mimetic and not a more potent version of phospho-mutant TA? The claim that TD is ER-retained should be revisited.

a) We have added new text in the Results section, accordingly to reflect that we cannot rule out the presence of non-glycosylated forms of beta subunit on the surface. However, when we compare the glycosylated vs non-glycosylated beta subunit we can clearly observe that glycosylated form is highly enriched on the surface when compared to input. This is in contrast to TA/TD phosphomutants.

b) We believe the TD signal that is observed at the surface fraction is background and is due to our technical limitations. We explained this in the existing text “As TD is likely sequestered in the ER, the TD bands observed (YFP- β 1/TD 15 ± 1.4 ; YFP- β 2/TD 46.1 ± 4.7 ; Fig S4A) suggest insufficient washing to remove residual biotin prior to lysis, which may indicate a further reduction of TA at the surface than observed.”

We believe that the diffuse TD signal that is observed in the dendrites is due to its ER retention as ER is present throughout the dendrites in neurons. In addition, we would like to point to the striking co-localization of TD with

ER marker expression in neurons in Fig S5B. We believe these provide strong evidence for ER localization of TD. ATP1a3 TA and TD neuronal expression show completely distinct patterns; we never observe TD in endosomal puncta and similarly we never observe TA in diffuse pattern in dendrites. The fact that there is no overlapping features between TA and TD localization strongly suggests that TD acts as a phospho-mimetic. Although phosphomimetics are known to act as phosphomutants in most cases, in this case our evidence support very distinct consequences of TA and TD mutations.

3) What is the evidence that NKA is recycled? Because authors observe co-localization of TA mutants with various compartment of endosomal pathway, it cannot be assumed that entire NKA trafficking pathway is dependent on this-specific phosphorylation as in the last Figure.

There are numerous phosphorylation sites in ATP1a3 as well as the beta subunits, in addition, binding proteins could regulate trafficking. We do not propose that the phosphorylation site we identified is the only one regulating the NKA trafficking. Our model is a simplification in this respect. We now added the word “simplified” in the last Figure legend. Recycling of NKA has been demonstrated in previous studies, recently in COS-1 cells. We have now included this new citation (Kristensen et al) in the Discussion section.

December 10, 2018

RE: Life Science Alliance Manuscript #LSA-2018-00118-TRR

Dr. Sila Ultanir
The Francis Crick Institute
1 Midland Road
London, London NW1 1AT
United Kingdom

Dear Dr. Ultanir,

Thank you for submitting your Research Article entitled "Chemical Genetic Identification of GAK Substrates Reveals its Role in Regulating Na⁺/K⁺-ATPase". I appreciate the introduce changes and it is a pleasure to let you know that your manuscript is now accepted for publication in Life Science Alliance. Congratulations on this interesting work.

DISTRIBUTION OF MATERIALS:

Again, congratulations on a very nice paper. I hope you found the review process to be constructive and are pleased with how the manuscript was handled editorially. We look forward to future exciting submissions from your lab.

Sincerely,
